# Plagued by the Past, Pressed by the Present: A One Health Perspective on *Yersinia pestis*

**DOI:** 10.3390/biomedicines13102555

**Published:** 2025-10-20

**Authors:** Andrea Ciammaruconi, Maria Di Spirito, Chiara Pascolini, Filippo Molinari, Orr Rozov, Marzia Cavalli, Giulia Campoli, Nathalie Totaro, Elisa Recchia, Silvia Chimienti, Anella Monte, Ferdinando Spagnolo, Florigio Lista, Raffaele D’Amelio, Silvia Fillo

**Affiliations:** 1Defense Institute for Biomedical Sciences, 00184 Rome, Italy; andrea.ciammaruconi@persociv.difesa.it (A.C.); maria.dispirito@persociv.difesa.it (M.D.S.); chiara.pascolini@persociv.difesa.it (C.P.); or maria.dispirito@uniroma1.it (M.C.); or giulia.campoli@dottorandi.unipg.it (G.C.); nathalie.totaro@persociv.difesa.it (N.T.); or elisa.recchia@alumni.uniroma2.eu (E.R.); or silvia.chimienti@students.uniroma2.eu (S.C.); or anella.monte@students.uniroma2.eu (A.M.); ferdinando.spagnolo@esercito.difesa.it (F.S.); florigio.lista@esercito.difesa.it (F.L.); 2Dipartimento di Sanità Pubblica e Malattie Infettive, Sapienza Università di Roma, 00185 Rome, Italy; 3Independent Researcher, 00153 Rome, Italy; 4Dipartimento di Chimica, Biologia e Biotecnologie, Università degli Studi di Perugia, Piazza dell’Università, 1, 06123 Perugia, Italy; 5Dipartimento di Scienze e Tecnologie Chimiche, Università di Roma Tor Vergata, 00133 Rome, Italy; 6Dipartimento di Scienze Cliniche e Medicina Traslazionale, Università di Roma Tor Vergata, 00133 Rome, Italy; 7Independent Researcher, 00162 Rome, Italy; raffaele.damelio@gmail.com

**Keywords:** *Yersinia pestis*, plague, pandemic, One Health, vaccines, antibiotics, monoclonal antibodies, bacteriophages

## Abstract

*Yersinia pestis*, the causative agent of plague, is arguably the most devastating pathogen in human history. Paleogenomic studies indicate its presence as early as the Neolithic era. It evolved from *Yersinia pseudotuberculosis*, with divergence estimates ranging from 1500 to 20,000 years ago, most often placed around 5000 years ago. Its natural reservoirs are wild mammals, particularly rodents, with fleas serving as vectors, while humans are incidental hosts. Over time, *Y. pestis* has acquired multiple virulence factors that disrupt immune responses and can lead to rapid, often fatal disease. Because the bacterium is maintained in wildlife cycles and can spill over to domestic animals, eradication is difficult, if not impossible. Nevertheless, mitigation is achievable using a One Health approach integrating human health, animal health, and the health of the environment. Neither vaccines nor monoclonal antibodies are currently licensed in most Western countries, thus, antibiotics remain the mainstay of therapy. Timely administration, ideally within 24 h of symptom onset, is critical, particularly in pneumonic forms. Phage therapy is under investigation as a potential treatment. Though often neglected in high-income settings, plague remains endemic in several regions, with the highest burden reported in Madagascar and the Democratic Republic of the Congo.

## 1. Introduction

Plague is a severe zoonosis, or infectious disease transmissible between animals and humans, caused by the highly virulent bacterium *Yersinia pestis.* The pathogen circulates among mammalian hosts, particularly rodents, which serve as a reservoir, as they are infected but generally resistant to the disease, and *Y. pestis* can be transmitted to humans through the bite of an infected flea [1]. Plague has been recognized for centuries and has caused three major pandemics and several minor epidemics with high lethality.

These outbreaks profoundly and negatively affected the economic and cultural development of large areas in Asia, Europe, Africa, and the Americas. In the 14th century, during the second pandemic, known as the “Black Death”, plague is estimated to have killed at least one-third of the European population [2]. Today, plague eradication remains unlikely, given that it is still present in over 26 countries worldwide, circulating among over 30 flea vectors and more than 200 mammalian host species [3], although the exact number of susceptible species remains unclear. Its control, therefore, can only be achieved through an integrated multi-sectoral One Health approach (Figure 1). While traditional health surveillance focuses solely on humans, understanding vector-borne diseases and their complex relationship to climate change requires an integrated and multidisciplinary approach [4]. The World Bank highlights the importance of integrated surveillance between human and animal health sectors, emphasizing how early disease detection in vectors or animals can prevent significant health costs [5].

The aim of this review is to provide a comprehensive and updated overview of *Y. pestis*, with particular focus on the One Health model and recent advancements in whole genome-based typing, diagnostic technologies, and countermeasure strategies against its potential use as a biothreat agent [1]. Recent innovations in whole-genome phylogeny and the elucidation of virulence mechanisms have helped bridge the gap between the classical microbiological approach and modern genomic and translational research. Special attention is given to innovative methods for pathogen detection, epidemiological surveillance and strategies for plague treatment and prevention including vaccine development, phage therapy, and biotechnology-based tools. By integrating historical, microbiological, and applied biomedical perspectives, this review highlights emerging approaches for mitigating the threat posed by *Y. pestis*.

## 2. One Health: Humans, Animals, and the Environment

The One Health approach is an interdisciplinary and systems-thinking-based paradigm that recognizes the interconnection between human health, animal health, and the health of the environment, representing a transition from a sectoral and isolated management of health issues to an integrated, collaborative, and systemic approach. It encourages collaboration among diverse disciplines such as human medicine, veterinary medicine, biology, environmental science, sociology, and psychology. The main objectives of this approach are to prevent, monitor, and control zoonotic and emerging diseases, reduce antimicrobial resistance, manage food safety risks, and integrate environmental protection into public health strategies [6]. Achieving these aims requires collaboration among scientists and policymakers. Key public health elements such as water, sanitation, and hygiene (WASH) are central to this approach. Poor management of human and animal waste can contaminate water and soil, encouraging heavy use of broad-spectrum antibiotics, thus facilitating the spread of antimicrobial resistance. Moreover, heavy rainfall events associated with climate change increase the risk of water source contamination with zoonotic pathogens, thereby increasing epidemic potential [7]. To address emerging health challenges, experts from multiple sectors have developed integrated surveillance and response systems (iSRS) for human and animal health [6]. After the coronavirus disease (COVID-19) pandemic, and following a joint initiative by the French and German Ministries for Foreign Affairs presented at the Paris Peace Forum in November 2020, four international agencies—the Food and Agriculture Organization (FAO) of the United Nations, the World Health Organization (WHO), the World Organisation for Animal Health (WOAH, formerly OIE), and the United Nations Environment Programme (UNEP)—established the One Health High-Level Expert Panel (OHHLEP) in 2021. This interdisciplinary body was tasked with advancing the transition of One Health from a conceptual framework to an actionable, integrated approach, grounded in the so-called “Four Cs”: Communication, Coordination, Collaboration, and Capacity Building [8]. Integration of human and animal surveillance, combined with the use of modern digital communication technologies, can significantly improve the timeliness of monitoring systems and enable rapid identification of outbreak-causing pathogens [9]. This emerging discipline enables the early detection of critical health events, outbreak forecasting, and timely alerts to public health or relevant authorities [10]. The main challenge lies in the availability of relevant data, as well as in its aggregation and analysis. Developing standardized reporting protocols is crucial to ensure data quality and reliability, enabling cross-sectoral experts to assess and compare methods and findings through a One Health lens [11].

### 2.1. Animal Factors

#### 2.1.1. Veterinary medicine and *Yersinia pestis*—A One Health approach

*Y. pestis* is of critical importance in veterinary medicine, as it is primarily maintained in wildlife reservoirs [12], with the capacity to spill over to domestic animals and humans [13]. *Y. pestis* has significant ecological impacts [14], causing large-scale die offs in populations that are susceptible [15], leading to destabilized ecosystems with reduced resilience. Plague is a major wildlife conservation concern, particularly for threatened species susceptible to depopulation [16]. Veterinary professionals, working in collaboration with public health authorities, conservation biologists, and ecologists, are crucial in surveillance, diagnosis, and mitigation efforts within wild and domestic animal populations [17]. Globally, *Y. pestis* circulates primarily in wild rodents, which act as the main reservoir and maintain enzootic transmission cycles [18]. The actual host range, however, is extensive, as nearly any mammal can be infected by the bacterium (Table 1). Over 350 distinct mammalian species have been identified as infectable by the bacterium; over 270 of these species are rodents [14]. *Y. pestis* can kill most susceptible mammals, with the disease manifesting across a spectrum, broadly characterized as enzootic or epizootic. Epizootic plague involves rapid, high-mortality outbreaks in highly susceptible species, often referred to as amplifying species. Enzootic plague involves slower transmission rates with lower or less obvious mortality over longer periods, sometimes involving more resistant hosts. Veterinary epidemiologists are crucial for understanding and interrupting these transmission cycles [14].

#### 2.1.2. Rodents and Other Sylvatic Reservoirs

Extreme environmental events, such as wet winters followed by cool summers, or floods, can drive rodent populations into human settlements, increasing the risk of disease transmission [19].

The reservoir systems in various foci are primarily composed of rodents, although other mammals also participate; humans are incidental hosts, as they develop severe bacteremia very rapidly and tend to die within a very short time [20,21]. Rodents are the primary reservoirs, followed by carnivores and lagomorphs [12]. Some wild birds, belonging to the order Passeriformes, can also be infected by plague. Carnivores are at high risk of infection; they are highly mobile and primarily feed on small mammals, namely rodents and lagomorphs. However, rodents play a fundamental role in spreading *Y. pestis* among different reservoir hosts, preventing its local extinction, and providing refuge for its survival, as they are both numerous and diverse across most *Y. pestis* ecosystems [22,23,24,25]. They include 2552 living species, accounting for 39.3% of the total mammal species richness [26]. Their ability to adapt to various lifestyles (e.g., arboreal, subterranean, semi-aquatic), their presence across a wide range of habitats (e.g., urban, desert, and wild habitats), along with their high reproductive rates and fast population turnover, make them highly effective hosts for the spread and amplification of zoonotic pathogens [27]. The two most important commensal rodent species, namely those living in close contact with humans and directly involved in the transmission of *Y. pestis*, are *Rattus rattus* (roof rat) and *Rattus norvegicus* (Norway rat), which act as reservoirs for the urban form of plague, while wild mammals are the reservoirs for the sylvatic form of plague [28]. In some rodent species, as well as in a few other host animals, the infection can persist over time, contributing to the maintenance of the pathogen in the environment. Wild rodents tend to show some resistance to the disease and therefore often do not develop severe clinical signs. In contrast, peridomestic rodents, such as rats living near or inside human dwellings, are generally much more susceptible to infection; once infected, they develop acute disease with high mortality rates [1].

#### 2.1.3. Vectors of *Y. pestis*

Fleas (order *Siphonaptera*), especially from the *Pulicidae* family, such as *Xenopsylla cheopis* (oriental rat flea) and *Pulex irritans* (human flea), are the main vectors of *Y. pestis*. The common louse, *Pediculus humanus* (family *Pediculidae*), has also been described as a vector [29].

Petrov [30] considered the term “vector” to be imprecise, as it does not fully describe the role of fleas in plague spread. This term overlooks a crucial aspect: the plague bacterium can survive in fleas much longer than in warm-blooded animals. This prolonged survival is essential during periods when infection drastically declines or ceases altogether, such as during the long hibernation of marmots (which lasts about nine months each year). For this reason, in the classification proposed by Burdelov [31], fleas are not merely considered vectors, but also “preservers” and “accumulators” of the plague microbe.

In many cases, flea infestations occur simultaneously on livestock and companion animals, which act as reservoirs and/or primary hosts [32].

The flea’s life cycle is influenced by environmental factors such as temperature, humidity, and the presence of a suitable host [32]. Some flea species live in nests or animal burrows in their adult form, while others remain on the host’s fur [32]. Their habitat preferences are influenced by factors that impact the survival of both adult fleas and their immature stages. The presence and behavior of potential hosts play a key role in shaping flea distribution and influencing their contribution to the spread of parasites and pathogenic bacteria [32]. Fleas live in close association with their hosts and rely on small but frequent blood meals to survive. Historical data provide insight into flea adaptability under different environmental conditions. For example, during the Black Death, the harsh climatic conditions of Northern Europe, characterized by cold and humidity, likely limited the presence of *Rattus rattus*, black rats, likely ruling them out as the primary host in those regions [33]. Conversely, fleas may have been mainly involved in pathogen transmission and maintenance even in Europe during similar climatic periods [34]. The flea is a true biological host, within which half of the pathogen’s life cycle occurs. *Y. pestis* remains localized within the flea’s digestive tract for the entire duration of its infection in the insect [35]. Two main mechanisms have been identified for the transmission of *Y. pestis* by fleas. The first, which takes place during the initial week after the flea becomes infected, involves the transmission of bacteria during the flea’s next blood meal. This process was once referred to as “mass transmission”, a term that underscored the observation that disease typically only arises when several fleas feed on an uninfected host at the same time [35]. More recently, this has been redefined as “early-phase transmission” [35].

A second transmission mechanism was described in 1914 by British medical entomologist Arthur W. Bacot [36]. According to this model of “late-phase transmission” or “biofilm-mediated transmission”, the process is driven by the formation of a dense bacterial biofilm by *Y. pestis* on the proventricular valve at the front of the flea’s midgut [37]. As the biofilm develops and consolidates, it gradually interferes with the proper closure of the valve and can eventually block it completely. When this happens, the flea attempts to feed, but the blood meal cannot enter its digestive tract, resulting in regurgitative transmission. The blocked flea becomes increasingly starved, which leads to more aggressive feeding behaviour and, consequently, a higher probability of transmitting the bacteria. It is notable that fleas often die while still attached to the host, suggesting that transmission might even occur after the flea’s death [35]. There is a common misconception that effective transmission requires the flea to be fully blocked. However, Bacot himself pointed out that fleas with only partial blockage are in fact more efficient at transmitting the infection than those that are completely blocked [38]. Indeed, fleas that are partially blocked have been observed to have a shorter extrinsic incubation period, a longer lifespan, and the ability to transmit infection with greater efficiency, even if they are not fully blocked.

#### 2.1.4. Animal: Epizootic and Enzootic Plague

Epizootic plague, frequently observed in prairie dogs due to their high vulnerability, describes a sudden outbreak that results in the death of at least 90% of the local host population within a limited geographic area and over a brief timeframe, usually just a few months’ duration [14].

In contrast, enzootic plague refers to a slower, less lethal form of *Y. pestis* transmission, where only a smaller proportion of animals are affected, the spread is limited to smaller areas, and the process unfolds over a longer period [39]. The mechanisms by which *Y. pestis* survives during enzootic phases, as well as the factors that cause it to shift to epizootic outbreaks, are still not fully understood and remain under debate [20,40].

Four main hypotheses have been proposed to explain these dynamics. The first hypothesis posits that the entire rodent community constitutes the natural reservoir of plague. The second hypothesis suggests that specific populations serve as true reservoirs, maintaining the infection over time and continuously infecting vectors. The third hypothesis proposes that fleas themselves represent a biological reservoir system, rather than mere vectors. Empirical data show that various flea species can survive in a blocked state for up to 558 days without feeding. After a period of dormancy or latency inside burrows, these fleas may resume feeding and infecting rodents, thus reactivating the transmission cycle. The fourth hypothesis proposes that *Y. pestis* may survive in contaminated soil, possibly by parasitizing soil protozoa or colonizing plant tissues. Additionally, the metapopulation model has been proposed to explain plague persistence [20]. According to this model, subpopulations exhibit a certain degree of isolation, allowing for the emergence of localized outbreaks affecting one or more subpopulations. After a population reduction or decimation in a specific area, neighbouring subpopulations can recolonize the depopulated zone [20]. This model explains the persistence and circulation of *Y. pestis* even in the absence of resistant hosts, likely facilitated by host movement between subpopulations [20]. *Y. pestis* could be maintained through sustained transmission between partially resistant enzootic hosts and their fleas with occasional spillover into more susceptible epizootic hosts, allowing for the amplification of *Y. pestis* and its epizootic spread or sustained enzootic transmission.

### 2.2. Environmental Factors

Climate change is among the most pressing challenges facing global health and the environment, deeply affecting various aspects of daily life. It refers to long-term shifts in climate patterns, primarily driven by anthropogenic activities such as fossil fuel combustion and deforestation [41,42]. Moreover, extreme weather events, including hurricanes, storms, floods, and droughts, are becoming more frequent and intense, thus destabilizing ecosystems. One significant consequence is the geographic expansion of infectious diseases [43], as higher temperatures expand the habitats of vectors such as mosquitoes and fleas, facilitating the spread of diseases like malaria, dengue, chikungunya, Zika, plague, and Lyme disease, even into temperate regions previously unaffected [44,45]. Climate variations influence the biological cycles of pathogen-carrying animals, potentially altering vector behaviour and migration, thus increasing the densities of the rodents and [46] the likelihood of epizootics with high pathogen transmission [44]. These ecosystem alterations create ideal conditions for the emergence and re-emergence of zoonotic diseases. In this context, *Y. pestis* remains a paradigmatic example highlighting the delicate evolutionary balance between rodent reservoirs and flea vectors, both strongly depending on ecological variation. Mild winters and humid early springs are associated with an increase in both human and animal cases, with a slight delay between environmental change and case occurrence [43]. Studies have reported conflicting results regarding the relationship between precipitation and the rate of plague transmission, likely due to regional differences and variations across temporal and spatial scales [47,48]. While rodents, the main reservoirs of plague, show behavioral plasticity, allowing short-term adaptation to climate fluctuations, extreme and prolonged environmental conditions (e.g., cold and drought) can exceed their adaptive capacity, facilitating the spread of the disease [34]. As early as the first half of the 20th century, Rogers hypothesized that seasonal variations in temperature and humidity influenced seasonal patterns of plague in India [49]. In Vietnam, plague incidence increases during the hot and dry season when preceded by heavy seasonal rains [50]. The relationship between temperature and the spread of plague is nonlinear, as temperature affects the pathogen, vectors, hosts, and human behavior [51]. Davis highlighted a lower incidence of plague in Africa during extreme temperatures, both high (>27 °C) and low (<15 °C) [52]. In the United States, flea-borne transmission was found to be more efficient at 10–15 °C compared to 23 °C, indicating that in colder regions, rising temperatures may reduce transmission [53,54]. In contrast, in Madagascar, plague spreads more effectively at higher temperatures, around 21 °C, because flea development, particularly *Synopsyllus fonquerniei* and *Xenopsylla cheopis*, is positively correlated with temperature [55]. Fleas, being ectothermic, are highly sensitive to temperature and humidity: survival decreases in hot, dry conditions, but increases in warm and humid environments [44]. These data highlight that temperature effects on rodents are less direct, since they are homeothermic and do not respond as rapidly to environmental changes [47]. Not only extreme environmental events, such as wet winters followed by cool summers, or flooding, but also regular seasonal changes, may alter rodent behavior by reducing available resources and driving them into urban areas thereby increasing bacterial transmission [19,56]. The relationship between environmental factors, hosts, and vectors is nonlinear, and small changes can lead to unpredictable outcomes [57]. Risk models identify two main hypotheses regarding the evolution of outbreaks [57]: the biodiversity amplification effect, suggesting that greater rodent diversity may maintain *Y. pestis* in the focus, and the trophic cascade hypothesis, which posits that climatic anomalies, such as high precipitation, can alter ecological dynamics by increasing food availability and the abundance of hosts and vectors, thus favoring pathogen transmission. Support for these models has increasingly been found across systems and points to a view of plague risk in which weather conditions (and their impact on flea vectors) in rodent biodiversity hotspots are the main drivers of transmission and spillover events.

In all endemic areas, human and animal cases are concentrated in a specific “plague season”, which is locally stable but varies between regions, including within the same country. For example, in Madagascar, the epidemic season runs from October to March in the highlands and from July to November along the coast. This variability is tied to favorable environmental conditions for rodents and fleas [58].

Like temperature, forest and pasture cover show nonlinear relationships with the rate of plague spread. Areas with extensive pastures or forests are associated with a higher risk of plague transmission, likely due to the limited access to effective healthcare systems compared to urban or agricultural areas. Moreover, developed transport infrastructure (road or river) facilitates disease spread. This pattern is observed in Europe, North Africa, and Southeastern America, where plague spreads faster compared to regions with less developed transport systems, such as South and Southeast Asia [57]. Increasing altitude was linearly associated with the increased probability of plague, while dense vegetation at lower elevations might hinder their presence of hosts. However, higher elevation can also limit food resources, thereby affecting host survival [59].

### 2.3. Human Factors

Contrary to common belief, the *Y. pestis* tends to spread more rapidly in low population density regions than in densely populated areas. This may be explained by more effective public health interventions in more urbanized regions, as well as by the fact that in less populated zones, transmission often occurs over long distances through trade and human mobility. Despite a higher number of cases in urban contexts, the rate of spread is slower [51].

Over time, the spread of *Y. pestis* has decreased, likely due to improved prevention and treatment strategies [51]. Human land use and environmental changes, such as landscape fragmentation, increase contact between vectors and hosts in rural and suburban areas with limited public health infrastructure [43]. A reduction in ecological diversity tends to favor species that act as reservoirs or amplifiers of the disease, increasing transmission risk [60]. Specifically, small mammals attracted to crops and waste can bring the sylvatic plague cycle closer to human environments [61] a phenomenon called the “inverse dilution effect” [62]. Natural barriers can limit *Y. pestis* spread, but human mobility continues to facilitate plague propagation by unintentionally transporting infected rodents, fleas, and infected individuals [43].

Social factors such as wars, famines, migration, poverty, inadequate sanitation, insufficient control of rodents and parasites, substandard housing quality, limited public health education, and social inequality increase plague risk by lowering hygiene standards and increasing human-rodent contact [34]. In plague-endemic areas, communities often have low perception of plague risk, which can hinder effective prevention and control efforts [63]. On the other hand, cultural behaviors—such as hunting, preparation and consumption of rodents, funeral and burial customs, and beliefs in supernatural causes for plague—could influence the spread of the disease [64,65,66]. Also, economic or political disruptions reduce public health services and surveillance, exacerbating plague outbreaks [64].

### 2.4. Plague Impact Assessment and Prediction

Over time, the need for reliable tools to assess and predict the impact of plague outbreaks has emerged. The problem is intrinsically complex and multidimensional: outbreak analysis depends on multiple biological (hosts, vectors, population ecology) and non-biological (climate, land use, socioeconomic structures) factors that modulate the intensity of transmission. The increasing availability of multidimensional data has fostered the development of mathematical models aimed at describing and predicting the impact of plague under different scenarios. Some general properties, emerging from the use of models, help evaluate their overall performance. First, scientific demand and data scale matter: recent studies conducted in the USA and China benefit from structured annual series of human and/or faunal cases, suitable for quantitative inferences and robust validation [67,68,69]. In contrast, pre-industrial Europe allows exploring low-frequency variability through paleoclimate proxies and historical records, but with more noise and marked non-stationarity [34]. Regarding the adequacy of the methods: Generalized Linear Models (GLMs; including logistic regression) perform consistently on rare event series [67]; logistic regression has also been used to estimate colony-level extinction probability in prairie dogs [70]. Multiscale analyses clarify spatial and temporal synchronies but remain mostly descriptive [68]; Generalized Additive Models (GAMs) capture nonlinearities in relationships [69]; Structural equation models (SEMs) decompose food pathways and feedback loops along the climate → vegetation → rodents → fleas → plague chain [71]; wavelet and coherence analyses isolate lags and synergies at interannual scales [68]; Bayesian additive regression trees (BARTs) add predictive power and allow the attribution of climate signals to observed patterns [57]. One limitation to the universal adoption of a single model across all plague scenarios is the heterogeneous availability of data. For example, it is possible to trace historical plague trends using climate annotations (temperature/humidity as proxies) [34] or distinct ecological regimes [69]. Where data are large, statistical associations consistent with trophic availability, flea physiology, and environmental constraints can be estimated, including colony context and landscape characteristics (e.g., topography and soils) in prairie dog systems [70]. However, uncertainties and limitations of external validity remain: rarity and underdiagnosis of human cases [57,67]; limitations of environmental proxies and some habitat covariates in specific contexts [59]; intrinsic uncertainties of historical proxies [34]; socioeconomic confounding in old series [34]; and restrictions on generalizability across different ecosystems [69,71].

The current research is supported by a broad spectrum of data, ranging from satellite observations to genomic information, which are employed in multiscale hierarchical models to combine global indices with local variables [34,57,68,71,72,73]. Such modelling approaches provide valuable tools to establish causal attribution, evaluate the influence of environmental matrix composition, and investigate host–vector network dynamics, enhanced by the feedback of empowered surveillance to parameterize and calibrate models, thereby enabling and improving predictive accuracy.

## 3. Epidemiological Surveillance and Preventive Strategies

In 2024, the WHO confirmed the inclusion of *Y. pestis* in the updated list of emerging and re-emerging pathogens with pandemic potential [74,75,76]. Following the severe pneumonic plague outbreak in Madagascar in 2017, the WHO published guidelines to improve global preparedness, emphasizing the need for differentiated approaches between endemic and non-endemic regions [1].

In endemic areas, cases are expected and managed by local health authorities, whereas in non-endemic regions, every case represents a public health threat, requiring immediate investigation and the implementation of control measures. In both contexts, the International Health Regulations mandate notification to the WHO within 24 h in the event of severe cases with a significant risk of international spread. Plague management is structured around four main components: collaborative surveillance and health intelligence, community protection, clinical care, and emergency coordination [1].

Epidemiological surveillance is essential to rapidly identify outbreaks and implement effective preventive measures, and it is crucial that this surveillance is integrated, collaborative, and multi-sectoral. It should be based on a One Health approach that monitors not only human and animal health, but also vector control. Therefore, collaboration among human, veterinary, and environmental authorities and health agencies is important to understand the complex interactions influencing plague dynamics [1,77].

Surveillance in animals, particularly through monitoring epizootic activity in the main reservoirs and vectors of *Y. pestis*, is fundamental [1,78]. Although rodents and their fleas are considered central to transmission, the exact number of species involved remains unclear [15,17]. The presence of domestic animals, such as dogs and cats, adds complexity to the epidemiological landscape, as they act as “bridges” between wildlife and human environments, facilitating pathogen transmission and sometimes serving as sentinel species [17,73]. Their role, however, and the ecological dynamics between natural reservoirs and incidental hosts, are difficult to monitor consistently, being influenced by ecological and behavioral factors that vary geographically [55,79]. In this context, seroprevalence among mammals, including domestic carnivores and other animals, can be useful for predicting new outbreaks [80,81]. In the USA, for example, the particularly monitored sentinel host is the coyote, whose natural habitat is the western USA, where plague is present. In a large serosurvey on over 25,000 animals, including carnivores (the species *Canidae*, *Felidae*, *Mustelidae*, and *Procyonidae*) and rodents, conducted between 2005 and 2010, the highest seroprevalence for plague was found in *Canidae*, *Felidae*, *Mustelidae*, with nearly all (98%) of the positive cases in *Canidae* represented by coyote, thus justifying its relevant role as sentinel species [71]. Although plague cases in the USA are sporadically observed in rural environments and are, therefore, hardly preventable, surveillance remains important for scientific purposes.

The scenario is different in China, where active plague foci are present, particularly in certain areas of Qinghai Province and the Tibet Autonomous Region, where the primary host is the Himalayan marmot, and in the southern Yunnan Province, where the primary host is *Rattus flavipectus*. This last focus, reactivated after 20 years of inactivity, currently accounts for 50% or more of the annual human plague cases in China. Cases occur in villages and urban environments, while in the first focus they are reported in rural areas as a result of activities such as hunting, skinning, or consuming marmots [71]. In Madagascar, shortly after the 2017 outbreak, a careful serological survey of mammals was conducted in five different districts of the island, selected based on the level of plague endemicity [82]. The seroprevalence in small mammals was 4.5%, and the highest flea index, the average number of fleas per rodent, was 8.8, found in Antananarivo, the capital. Another study demonstrated the usefulness of using dogs as sentinel animals for plague in endemic outbreaks in Madagascar [83].

In light of the epidemiological behavior of plague, which has re-emerged in India after 30 years [84], in Algeria after more than 50 years [85], and in Libya after 25 years [86], analytical sero-surveillance studies have also been conducted in countries without reported human plague cases for a decade, such as Zambia [87], or for even longer periods, as in Iran [18].

Endemic areas show great ecological variability, which affects the distribution of vectors and reservoirs. From equatorial forests to the arid regions of Central Asia, geographic conditions shape transmission dynamics and complicate the adoption of uniform surveillance strategies [1,15,17]. While fleas are generally regarded as the main vectors, in some areas such as Madagascar, other ectoparasites contribute to the persistence of the disease, further complicating monitoring efforts [55]. Therefore, surveillance strategies must account for these regional variations by using different indicators. Epidemiological surveillance is a complex process that combines both active and passive activities [1]. Active surveillance involves continuous and targeted monitoring of rodent and vector populations, such as fleas, as well as the environmental conditions that favor their proliferation. For example, visual investigations of rodent mortality including reports of “rat falls” or sylvatic rodent die-offs can provide early warning signs of epizootic activity. “Rat falls” are defined as the death of multiple commensal rats in one or more dwellings, provided the deaths are not due to poisoning [1]. Continuous surveillance allows timely calculation of the host density, as reported in the plague manual of WHO, based on the ratio of animal caught/number of traps during a specific time period per surface unit; high rodent host density is an indicator of a high risk of epizootic episodes that may spill over to humans. In this case, rodents should be reduced, but this operation during epizootic periods should be preceded by the control of vectors with insecticides, in order to prevent a higher risk for humans, in case starving fleas have not enough available rodents for feeding, thus attacking humans [1,71].

Flea surveillance, although requiring careful analysis due to its relative imprecision, is also useful for detecting ongoing outbreaks [1]. This is conducted by calculating the flea index, which refers to the number of fleas specific to particular rodents. For *Xenopsylla cheopis*, for instance, a specific flea index greater than 1 indicates a significant disease risk, thereby recommending control measures. Plague prevention programs include controlling vector and reservoir populations through the use of insecticides and rodent control, measures that should be applied with caution to avoid disrupting ecological balance.

The most effective preventive strategies for limiting the transmission of plague from rodents should include reducing habitats near homes, protecting food supplies, minimizing contact with rodents and their feces, using protective equipment when handling potentially infected animals, implementing environmental modifications to discourage rodent colonization, and applying repellents and antiparasitic treatments to pets. Adopting these measures can significantly reduce the risk of infection while preserving ecological balance [88]. Complete eradication of rodents is, in fact, not only unfeasible, due to their widespread distribution, but also ecologically harmful, as rodents are insectivores and prey for larger predators. Attempts to vaccinate animal hosts have been carried out, with promising results [89]. Additionally, improper use of insecticides could lead to the development of resistance in fleas [90], while excessive rodent culling during epidemics may increase the risk of flea infestations [1,17]. Therefore, control measures should primarily focus on inter-epidemic periods, when the risk of spread is lower.

Passive surveillance, on the other hand, collects data from health sources to enable rapid response to outbreaks. International cooperation is essential to contain the spread of plague, which can quickly propagate through trade and migration flows [1,17].

Careful surveillance of plague in China, dating back to 1980, has allowed for the observation of the response to climate change in two different rodent-vector-plague systems The two analyzed foci are located in adjacent areas of Inner Mongolia, with one system having the Mongolian gerbil as the primary host and the other having the Daurian ground squirrel. By applying structural equation modeling (SEM), it was possible to observe a positive association between vegetation and the Daurian ground squirrel density, while the association between vegetation and the Mongolian gerbil density was negative, thus indicating that careful surveillance of hosts can enable tailored interventions based on the different responses observed [71].

The distribution of plague is influenced by numerous environmental and anthropogenic factors, including altitude, environmental changes, deforestation, agricultural expansion, urbanization, and the introduction of invasive species, all of which affect risk levels and geographic spread [17,91]. Interactions between rural areas, often characterized by poverty and poor sanitation, and densely populated urban areas with inadequate health services facilitate transmission, particularly of the highly contagious pneumonic form [92]. The 2017 outbreak in Madagascar highlighted how deficiencies in integrated surveillance, intersectoral communication/coordination and ineffective sanitation standards contributed to the spread of the pathogen in vulnerable urban settings [1,17,92].

Surveillance should also extend to phases of low epidemic incidence, supported by environmental and meteorological data that influence vector and reservoir populations. Climatic factors such as temperature, humidity, and precipitation affect flea survival and the seasonality of epidemics [64,78,81]. Climatic events such as El Niño have also been associated with an increase in epidemics in countries like Kazakhstan, Uganda, the United States, and Madagascar [55,68,93]. Careful plague surveillance carried out for 56 years in the USA allowed us to observe that the Pacific Decadal Oscillation (PDO), together with previous plague levels and above-normal temperatures, may jointly provide an explanation for the plague variability [67]. The hypothesis is that PDO influences the climate, by increasing temperatures and precipitation, thus, according to the trophic cascade theory, increasing the food availability, the abundance of hosts and vectors, and finally plague human cases. Even El Niño Southern Oscillation (ENSO), in combination with PDO, may influence the plague dynamics in the western USA [68].

Transmission is further facilitated by the movement of infected wild animals and the introduction of the pathogen into human environments via incidental hosts such as domestic animals, underscoring the need for integrated control strategies involving both human and animal populations [17].

Anthropogenic factors such as trade, migration, and agricultural activities alter local ecosystems and may increase epidemic risk [17]. Therefore, surveillance at international entry points (e.g., ports, airports) is also critical, this includes monitoring rodent and flea populations that pose a high risk of spread and require timely notifications of suspected or confirmed plague cases to WHO and relevant veterinary or animal health authorities for coordinated control measures [1].

Prevention demands an integrated approach, aligned with the One Health framework, that includes health education and effective surveillance. However, in low-income countries, the implementation of such interventions is limited by inadequate healthcare infrastructure, insufficient resources, and socioeconomic barriers.

Raising public awareness about the risks associated with contact with dead animals, flea prevention, and timely reporting of suspected cases is essential but often hindered by cultural barriers and limited access to information. Similarly, adequate hygiene practices, proper food storage, and the distribution of personal protective equipment are frequently insufficient [1,17]. Clinically, timely administration of antibiotics and protection of healthcare personnel during patient and corpse management are crucial [94]. Well-coordinated emergency responses among local authorities, health institutions, and international organizations, with adequately trained rapid response teams (RRT), enable effective and scalable interventions [1]. Close proximity between human populations and rodent reservoirs, excessive use of antibiotics in agriculture, and contamination from antibiotic residues in hospitals contribute to the selection of resistant strains, complicating therapies and increasing costs, especially in low-income settings [1,91,95]. Rapid diagnostic capacity and reference laboratories for genetic and molecular analyses are thus essential for timely detection and response to resistant strains [1]. Modern technologies such as genomics, bioinformatics, and artificial intelligence enhance monitoring and outbreak prediction.

This can be an opportunity for the vulnerable low- and middle-income countries, because the surveillance of pastoralists and their households and livestock may easily and economically be conducted by mobile phones [10].

The “Big Epidemiology” approach, which integrates large datasets of historical, genetic, and environmental data, can more effectively identify potential hotspots and guide targeted interventions [96].

Analysis of historical cases and studies on the coevolution of *Y. pestis* and human populations may help predict future outbreaks based on established models [97].

Plague remains a significant public health threat, with transmission dynamics influenced by environmental, animal, and human behavioral factors. Comprehensive integrated surveillance systems that monitor wildlife, domestic animals, and socio-ecological variables are necessary. The adoption of technological innovations and ongoing international cooperation are fundamental to containing the risks associated with this disease.

## 4. Epidemiology

Human cases of plague are mainly linked to the epizootic activity of natural foci. These natural foci are approximately located along the latitudes of 49°N in Eastern Europe, 34°S in Africa, 51°N in North America and East Asia, and 37°S in South America. The region spanning these latitudes can be described as the “plague belt” [98].

In general, across Africa, Eurasia, and the Americas, plague foci are concentrated in arid regions, deserts, semi-deserts, dry savannas, steppes, and montane grasslands that host characteristic communities of warm-blooded animals and their ectoparasites [98].

Natural plague foci are areas where the disease persists in animal populations and the environment, occasionally leading to sporadic human cases or even larger outbreaks, such as the significant one that occurred in Madagascar in 2017.

Although the number of reported human cases has been relatively low since 1945, plague remains a threat due to the vast areas with natural foci where the disease is endemic [99]. A total of 26,237 human cases of plague were reported to the WHO by 21 countries in the Americas, Africa, and Asia between 2000 and 2018 [29]. Between 2000 and 2019, 371 cases of plague were reported in the Americas, mostly as isolated cases in the United States and a few localized outbreaks in South America [29].

In contrast, Africa remains the continent most affected by plague, representing 97% of global cases [29], with 25,409 cases primarily documented in the Democratic Republic of the Congo and Madagascar. In 2017, a major outbreak occurred in Madagascar, with approximately 2400 cases, of which 78% were clinically classified as pneumonic plague [100]. However, only 32 cases were microbiologically confirmed [100]. In North Africa, Algeria reported 15 cases in 2003 [85], and 13 cases were documented in Libya, near Tobruk, in 2009 [86]. Molecular analyses identified two distinct strains: *Y. pestis* IP1860-64 biovar Orientalis in Algeria and *Y. pestis* IP1973-75 biovar Medievalis which is still circulating and was involved in isolated outbreaks, such as the 2009 Libyan case; it is not associated with the third pandemic [86]. In Mongolia, most cases of bubonic plague recorded in 2010 were attributed to marmots (75.2%) [101,102], a pattern also observed in Kyrgyzstan and Russia, where two cases were contracted by marmot hunters [103,104].

During periods when no human cases of plague are reported, it is difficult to determine whether the bacterium has truly disappeared, continues to circulate in animal reservoirs with interrupted transmission to humans, or whether human infections have gone unnoticed or misdiagnosed.

## 5. Microbiological Profile of *Y. pestis*

*Y. pestis* is a Gram-negative, non-motile, non-spore-forming, facultative anaerobic and intracellular bacillus or coccobacillus. *Y. pestis* belongs to the *Yersiniaceae* family within the order *Enterobacterales*, reflecting a taxonomic revision from the previous classification within the *Enterobacteriaceae* family. Molecular and genomic studies have revealed that *Y. pestis* evolved relatively recently from its ancestor *Y. pseudotuberculosis* [21,105,106], an enteric pathogen causing yersiniosis, a gastrointestinal disease. Despite sharing more than 97% nucleotide sequence identity with *Y. pseudotuberculosis* across 75% of the chromosomal genes [107], *Y. pestis* exhibits substantial differences in ecological niche, pathogenicity, and disease progression [106]. Evolutionary emergence of *Y. pestis* includes the acquisition of the virulence-associated plasmids pMT1 and pPCP1, which conferred the capacity for flea-borne transmission and increased tissue invasion [108,109,110]. This was accompanied by gene loss and functional inactivation due to deletions and single-nucleotide polymorphism (SNP) mutations [111] in genes associated with enteric colonization and motility. These processes reflect the transition from an enteric to a vector-borne invasive pathogen [106,109].

Among the members of the genus *Yersinia*, *Y. pestis*, *Y. pseudotuberculosis*, and *Y. enterocolitica* are the only species recognized as human pathogens. *Y. pestis* is a small bacterium (0.5–0.8 µm × 1–2 µm), relatively fastidious, growing on blood agar (BA), MacConkey agar (MAC), and other enriched media. Although *Y. pestis* can grow on these standard culture media, the use of a selective agar containing cefsulodin-irgasan-novobiocin (CIN) enhances its isolation from polymicrobial samples such as sputum.

Moreover, it shows a characteristic bipolar staining with aniline dyes, such as Giemsa or Wayson, particularly evident in fresh clinical specimens, producing the characteristic “safety pin” appearance [112]. The organism displays a broad temperature tolerance (4–40 °C), with optimal growth temperature between 28 °C and 30 °C. This reflects the evolutionary adaptation to the flea midgut environment, while growth at 37 °C is slower and associated with the expression of mammalian host-specific virulence factors [113]. *Y. pestis* proliferates optimally in a pH range between 7.2 and 7.6, although it can tolerate values from pH 5.0 to 9.6 [114]. Colony formation usually requires 24–48 h on enriched media, in ambient air, although incubation in 5% CO_2_ may enhance growth. In defined liquid media, the doubling time can be as short as 75 min under optimal conditions [21]. On agar plates, *Y. pestis* typically grows as grey-white, translucent, slightly mucoid non-hemolytic colonies, showing a raised, irregular morphology at 28 °C. A characteristic feature of *Y. pestis* in broth culture is flocculent or aggregated growth with the formation of “stalactites”, due to aggregation of the bacteria and adherence to the sides and bottom of the tube.

This is enhanced by the expression of specific virulence factors [115]. At all growth temperatures, *Y. pestis* shows auxotrophies for L-isoleucine, L-valine, L-methionine, L-phenylalanine, and either glycine or L-threonine. These metabolic characteristics reflect the presence of cryptic or pseudogenized genes in several biosynthetic pathways [116,117].

The microorganism also requires biotin, thiamine, pantothenic acid and glutamic acid, when cultured at 37 °C [21,118].

Moreover, *Y. pestis* is oxidase, indole and urease-negative, catalase positive and non-lactose fermenter [112]. Although it exhibits a significant dependence on host derived nutrients, which precludes a saprophytic mode of life, *Y. pestis* is capable of replicating extracellularly under laboratory conditions; therefore, it cannot be classified as a strict parasite. On the basis of geographical and biochemical characteristics, *Y. pestis* isolates can be classified into two subspecies: *Y. pestis* subsp. *pestis*, typically associated with human infections and comprising the biovars Antiqua (ANT), Medievalis (MED) and Orientalis (ORI) [108]; and *Y. pestis* subsp. *microtus*, primarily involved in zoonotic cycles and generally considered avirulent or of low virulence in humans [119,120,121,122]. Advancements in molecular typing methods and whole-genome sequencing of *Y. pestis* strains, collected from worldwide diverse foci, have significantly refined the global phylogeny of *Y. pestis* [123,124,125].

## 6. History and Evolution of *Y. pestis* and Plague

### 6.1. Phylogeny of Y. pestis over Time

Application of modern genomic techniques, particularly next-generation sequencing (NGS), enabled the complete reconstruction of *Y. pestis* genomes and provided valuable insights into its evolutionary history [126]. Extraction of ancient DNA (aDNA) from archaeological remains has provided insights into both historical human populations and ancient microbial and viral pathogens, thus helping to track the evolution of the pathogens [127] as well as entire microbiomes preserved in a variety of archaeological materials [128]. Notably, the typically low quantity, high degree of fragmentation, and extensive chemical damage present in aDNA samples pose significant challenges for their analysis, particularly in terms of reproducibility and authentication [129]. However, the high reproducibility of NGS technologies, along with their broad analytical capabilities—including SNP analysis, deletion mapping, whole genome sequencing (WGS), CRISPR (clustered regularly interspaced short palindromic repeats) analysis, and multiple-locus variable-number tandem repeat (MLVA) analysis—has allowed laboratories to reliably generate, integrate, and compare data across studies [129], thereby enabling the recovery of high-resolution genetic information.

This process allowed the reexamination of the founding hypotheses and the classifications underlying them.

It has been estimated that the *Yersinia* ancestor emerged approximately 42 to 187 million years ago, while *Y. enterocolitica* and *Y. pseudotuberculosis* diverged from their common ancestor between 0.4 and 1.9 million years ago [58,108,111] (Figure 2).

The acquisition of the pCD1/pYV, an important 70 kb virulence plasmid facilitating infection, marked a shift from a hypothetical non-pathogenic environmental *Yersinia* to a human-pathogenic species.

The enteropathogenic *Yersinia* species included *Y. enterocolitica*, *Y. pseudotuberculosis* and *Y. pestis* [130]. The pCD1/pYV plasmid encoded the *Yersinia* outer-membrane proteins (Yops), which constitute an array of effectors directly translocated into host cells through a type 3 secretion system (T3SS) (for an updated in-depth review of *Yersinia* T3SS and Yops see refs. [85,91]), also found in the enteric pathogens *Y. pseudotuberculosis* and *Y. enterocolitica*. The Type III secretion system (T3SS) exhibits a needle-like configuration on the bacterial surface. This structure facilitates the injection of Yop into host cells upon direct interaction between the pathogen and the host cells [91].

Thus, Yops can play a critical role in plague pathogenesis by counteracting the innate immune defenses of the host. Evidence from ancient genome sequencing and subsequent phylogenetic analysis has shown that *Y. pestis* diverged from *Y. pseudotuberculosis* approximately 1500 to 20,000 (5000) years ago [58,108,123,131].

Moreover, it was estimated that the most basal *Y. pestis* lineages derive from an early divergence event that occurred between 5000 and 5700 years ago. The event gave rise to both a lineage still persisting today (0.PE7 and 0.PE2) and to two extinct Neolithic and Bronze Age lineages (Gok2 and LNBA, respectively) [56,111,127]. This recent divergence led to highly clonal microorganisms, as confirmed by studies on *Y. pestis* genomic profile and highlighted by high genetic relatedness to *Y. pseudotuberculosis* [58]. For instance, 75% of chromosomal genes of CO92 *Y. pestis* strain share ≥97% nucleotide identity when compared to homolog genes of *Y. pseudotuberculosis* IP32953 strain [107].

In the evolution of *Y. pestis* from *Y. pseudotuberculosis*, both the inactivation of genes essential for an enteric lifestyle and the acquisition of two plasmids (pMT1 and pPCP1) harboring genes encoding new virulence factors were involved [58]. During evolution, *Y. pestis* lost 317 genes, while 208 genes deriving from *Y. pseudotuberculosis* became pseudogenes [107]. At the same time, *Y. pestis* acquired 32 genes, grouped into six genomic clusters with different characteristics; however, most do not appear to encode virulence factors [107]. *Y. pestis*’ evolution resulted in the loss of genes involved in intestinal adhesion and invasion; in parallel, the microorganism retained genes that facilitate host colonization, such as the heme uptake locus (*hmu*), and several chromosomal-encoded genes as the *ph6/psa* fimbrial operon and the attachment-invasion locus *ail* [132,133]. With regard to plasmids, one of the two unique plasmids acquired by *Y. pestis*, is a ~9.5 kb pPCP1/pPla/pPst plasmid, encoding a bacterial surface-bound protease (plasminogen activator, Pla), a key virulence factor that facilitates dissemination by its potent fibrinolytic activity [58].

It also contains genes encoding for pesticin and coagulase, which facilitate flea-borne transmission, overcoming host defences [58].

The pFra/pMT1 plasmid (100–110 kb) harbors genes encoding two major virulence factors: the phospholipase D and the Fraction 1 (F1) antigen. The *Y. pestis*-specific pFra/pMT1 plasmid was acquired through horizontal gene transfer [131]. The F1 antigen forms a polymeric antiphagocytic capsule around the bacteria contributing to immune evasion [58]. While the phospholipase D, also known as *Yersinia* murine toxin (Ymt), is not required for typical plague disease progression and virulence in mammalian hosts [127], its acquisition was a critical event in the evolution of *Y. pestis* and the transition towards an insect-borne life cycle. Ymt enhanced *Y. pestis* survival within the flea gut, greatly expanding its range of mammalian hosts and their associated fleas, and promoting efficient transmission between hosts [58,131]. These genetic adaptations highlight the pathogen’s dynamic nature and its ability to evolve in response to environmental pressures [133].

The biochemical characterization of the pathogen enabled the classification of strains into biovars [129], based on their ability to ferment glycerol and reduce nitrate. This approach allowed the phylogenetic classification of modern strains into three biovars (ANT, MED, ORI) [127,129].

In particular Antiqua reduces nitrate and ferments glycerol, whereas Medievalis does not reduce nitrate and Orientalis does not ferment glycerol, as if Antiqua were ancestral to both Medievalis and Orientalis [123]. It has been shown that each of the biovars seems to be distinct according to the genomic patterns of IS100 insertion elements [108].

These insertion sequences generate genomic instability and create a polymorphism of the hybridizing restriction fragments (restriction fragment length polymorphism [RFLP]), which can be used to subtype this relatively clonal species [134]. For this reason, despite the very high degree of conservation at the gene level, the genotypic subdivision of the species *Y. pestis* has been possible and the variations in the chromosomal location of IS100 have been used to study the microevolution of *Y. pestis* [134]. Achtman et al. reported distinct clustering of biovars in a phylogenetic tree based on the chromosomal locations of an insertion element [108]. The longer phylogenetic branches of Antiqua and Medievalis are consistent with the previous hypothesis of Devignat formulated on biochemical characterization [119], that these evolved earlier than Orientalis and with the assignment of biovar Antiqua to the first pandemic wave of human plague and biovar Medievalis to the second wave [108]. Moreover, based on multiple spacertyping (MST) [93] analysis of specific gene deletions, such as the 93 base pair deletion in the gene encoding glycerol-3-phosphate dehydrogenase (glpD) by sequencing from ancient dental pulp specimens, researchers confirmed the absence of glycerol fermentation in the Orientalis biotype, which was present in isolates of the other biovars [92,94]. Extensive phylogenetic studies focusing on the diversity of *Y. pestis* strains have enhanced our understanding of the bacterium’s diversity and evolutionary history, and revealed the presence of five major phylogenetic branches in *Y. pestis* [127]. Based on whole genome sequencing, a phylogeny including biovar IN = Intermediate (which reflects a transitional state from Antiqua to Orientalis) and PE = Pestoides (including Microtus isolates) was proposed (Figure 3) [111]. According to this classification, branch 0 represents the most ancestral lineage and includes strains distributed across China, Mongolia and the areas of the former Soviet Union, primarily belonging to the group known as “Pestoides” [127]. Several evolutionary lineages emerged from the ancestral branch, specifically branches 1, 2, 3, and 4. Medievalis (2.MED) is a sub-branch of branch 2, while Orientalis (1.ORI) forms a distinct lineage within branch 1. Regarding branch 1, sub-lineage ORI.1 is associated with the third plague pandemic that spread worldwide during the 19th and 20th centuries (Figure 3) [127]. This branch is still responsible for more confined epidemics such as those reported in Madagascar [127].

Branches 1–4 are currently widely distributed across Asia, Africa, and the Americas, probably originated from a rapid population expansion event named “Big Bang” [127]. This fundamental evolutionary event probably occurred between the 13th and 14th centuries CE, and preceded the spread of the medieval plague. It gave rise to several extant lineages within a short evolutionary time, leading to genetic radiation that likely enabled the extensive and rapid spread of *Y. pestis* during the Second Pandemic [125].

While the genealogy of *Y. pestis* has been clarified thanks to the advent of whole genome sequencing, the forces that shaped this process and the dynamics that drove its evolution have not been fully elucidated. As a result, using the One Health approach, which combines the study of genetic variation such as SNPs, indels, and gene gain/loss with the analysis of environmental factors such as climate, soil components, hosts, and vectors, would allow us to reconstruct the forces that shaped its evolution. This could lead to a better understanding of the natural survival strategies of *Y. pestis* and the dynamic forces that shaped its genome.

However, the studies of aDNA have indicated that, although a dynamic plasticity with rearrangements emerges [135], the *Y. pestis* genome is considered a monophyletic species. This is confirmed in an interesting study of aDNA extracted from the dental pulp of victims of the plague of London, 1348–1350, which showed a few mutations between this aDNA and the genome of currently circulating *Y. pestis* strains [136]. The perceived higher virulence of the microorganism in the Black Death, therefore, could not be explained by the presence of different virulence factors compared to the currently circulating strains. The different virulence may only be attributed to other components of the plague eco-system, such as environment [69], vector dynamics [137], and host susceptibility [138], according to the One Health approach.

### 6.2. A Historical Overview of Yersinia pestis and Its Impact on Human History Through the Plague Pandemics

Three plague pandemics have been historically recorded after the Bronze Age: the first which began with the Plague of Justinian in 541 CE, the second plague pandemic known as the Black Death and the most recent third plague pandemic which began in the mid-19th century [127].

In addition, numerous smaller epidemics and sporadic outbreaks, such as the Plague of Milan (1629–1631) [139,140], the Great Plague of London (1665–1666) [141], the Plague of Marseille (1720–1722) [142] and the Hong Kong Plague outbreak (1994–1995) [143] have been documented (Table 2); each of these outbreaks profoundly reshaped the affected societies, collectively resulting in tens of millions of deaths.

These outbreaks occurred within the broader context of the Second and Third Plague Pandemics, respectively, highlighting the prolonged and recurring impact of plague across centuries [144].

The first plague pandemic, known as the Justinianic Plague, named after the Byzantine Emperor Justinian I, occurred between the 6th and the 8th centuries CE. It most likely originated in East Africa, particularly in the area of present-day Ethiopia or Sudan, and spread northward to Egypt before reaching Constantinople in 541 CE. From there, the disease rapidly disseminated across the Mediterranean basin, reaching as far as Rome [133]. Its spread was facilitated by trade routes connecting Africa and the East, particularly along grain trade routes and through the movements of the Byzantine military [145]. The first outbreak occurred between 541 and 544 CE, but the plague did not disappear thereafter.

Instead, it reemerged in multiple phases over the following centuries, with recurrences estimated approximately every 15–20 years. Overall, between 14 and 21 epidemic waves are believed to have struck the Empire up to the late 8th century, establishing a nearly two-century pattern of endemic circulation [146]. The disease reached Rome and Western Europe, causing high mortality and contributing to the weakening of the Byzantine Empire. Estimates of mortality vary considerably, ranging between 15% and 40% of the population [21].

A recent study, based on historical and archaeological data, suggests that the death toll of the Justinianic Plague may have been overestimated.

The authors propose that the global mortality rate may have been as low as 0.1%, concluding that the plague likely had a limited demographic, political, and social impact [146].

The Second Plague pandemic, commonly known as the Black Death, represents a pivotal epidemiological and historical event, which paralleled—and in many ways surpassed—the Justinianic Plague in demographic and societal impact [136]. It likely originated in the late 1330s in Central Asia—possibly within the territories of present-day Kazakhstan, Southern Russia, or China—where the bacterium *Y. pestis* had established enzootic reservoirs [127]. The plague spread westward along well-established trade routes, both overland via the Silk Road and across the maritime networks of the Indian Ocean and Black Sea [124].

By 1346, the disease had reached the Crimean Peninsula, where it was reportedly introduced into Europe through the Genoese trading post of Kaffa (Feodosiya) during a Mongol siege, which some sources describe as an early example of biological warfare, through the catapulting of infected corpses over city walls [147].

Between 1346 and 1353, the Black Death swept through Europe with extraordinary speed and lethality, resulting in the death of an estimated 30% to 40% of the continent’s population [2]. In some regions, mortality may have been even higher.

Unlike the first pandemic, which established an endemic pattern within the Byzantine Empire, the second pandemic evolved into a transcontinental crisis with long-lasting consequences.

The Black Death marked the beginning of a prolonged plague epoch in Europe, with recurrent epidemic waves persisting until the 18th century. While the first pandemic’s demographic impact remains debated, the second pandemic is widely regarded as one of the most transformative crises in European history, triggering structural changes whose effects would shape the continent for centuries [148].

The third plague pandemic, spanning from the late 19th to the mid-20th century, likely originated in Yunnan province, China, around the late 18th century. It spread globally during the 19th and early 20th centuries, aided by the expansion of maritime and rail transport [29].

The disease reached various continents, including Asia, the Middle East, Africa, Oceania, Europe, and the Americas. Despite efforts to control its spread, the pandemic resulted in millions of deaths, including 60,000 fatalities in Hong Kong in 1894 [149]. During the Hong Kong epidemic of 1894, Alexandre Yersin and Shibasaburo Kitasato independently announced the isolation of the plague bacterium [21].

In 1897, during an outbreak in India, Masanori Ogata and Paul-Louis Simond independently confirmed the role of rats and their fleas as vectors of *Y. pestis*, marking a significant milestone in the study of vector-borne infectious diseases [150]. This discovery laid the foundation for modern epidemiological control strategies targeting zoonotic reservoirs and arthropod vectors [21]. The pandemic caused over 26 million cases and more than 12 million deaths in India and China. Unlike in the United States, where it became endemic, the plague did not establish itself permanently in Australia [21,133]. Today, the plague remains endemic in several regions, particularly in parts of Asia, Africa, and the Americas.

## 7. Pathogenesis, *Y. pestis* Virulence Factors, Escape Mechanisms, Host Immune Response, Clinical Forms

### 7.1. Pathogenesis, Y. pestis Virulence Factors, Escape Mechanisms, Host Immune Response

*Y. pestis* holds three plasmids: the first, pCD1 (calcium dependence), in the strain KIM (Kurdistan Iran Man), even designated pCad, pVW, pYV, or pLcr, of ~70 kilobases of DNA, carries the low calcium response (LCR) genes, which encode a type 3 secretion system (T3SS) and its secreted effector proteins [29]. They are produced and work at 37 °C, in the presence of low calcium concentration, and are represented by one V (virulence) antigen and other Yops, which are able to (1) block phagocytosis; (2) induce apoptosis; (3) modulate cytokine network to inhibit the inflammatory response [21,151,152,153].

The other two plasmids are pPCP1 and pMT1. pPCP1 (pesticin, coagulase, plasminogen activator), also designated pPst, pPla, or pYP, of ~9.5 kilobases of DNA, encodes for a plasminogen activator (Pla), that also works as adhesin, and bacteriocin pesticin (Pst). pMT1 (murine toxin), even designated pFra/Tox, pFra, pTox, or pYT, of ~100–110 kilobases of DNA, encodes for Ymt and capsular protein fraction 1 (CaF1), this last one equipped with adhesin and antiphagocytic activity [21,29,112,152,154,155]. All these factors allow the bacterium to enter the human host through the bite of the infected flea and via the lymphatic vessels to quickly travel to the draining lymph nodes [156], remaining invisible to the immune system, thus able to multiply undisturbed. While Ymt, which is a phospholipase D, acts principally to favor the bacterial survival in the flea midgut at 26 °C [157], the other virulence factors generally act to reduce the host immune response to the infection [158]. In addition to Ymt, the four proteins encoded by the hemin storage *(hms)HFRS* locus also act at 26 °C to favor the permanence of *Y. pestis* in the flea midgut, being involved in the production of biofilm matrix exopolysaccharide. This strengthens the resistance of *Y. pestis*, which is allowed to multiply and form an aggregate that fills the proventriculus, eventually blocking the flea midgut [106]. Once the blocked flea bites again to take its next blood meal, it is forced to regurgitate the bacterial aggregate, thus effectively infecting the new host [159]. The One Health approach is especially valuable for studying the genetic evolution of the *Y. pestis* virulence factors under the environmental pressure, as recently suggested [160]. Cui et al., by analyzing 78 isolates of *Y. pestis* collected during the period 1967–2006 from the Guertu natural plague reservoir in China, found evidence for a pressure in the RNA polymerase ω-subunit encoding gene *rpoZ*. Mutations in this genomic area were positively associated with climatic variations as colder and drier environments and also with a capacity of overproduction in biofilm formation in the gut of fleas. These associations are highly suggestive for a direct influence that the environmental/climatic pressure may exert on the genetic evolution of *Y. pestis*, by shaping and tuning its virulence factors. This observation opens up new horizons on the influence that the environment exerts on the plague eco-system, which was already well known, but limited to an indirect action through hosts and vectors [160].

Hms is present in virulent pgm+ (pigmentation positive) *Y. pestis* strains, which need iron that is transported by the siderophore yersiniabactin complex system, to grow and express virulence [21].

The need for iron for expression of virulence is underlined by a fatal case of human plague [161] reported in a laboratory researcher who was actively involved in the plague research by utilizing a pgm− (pigmentation negative) *Y. pestis* strain (KIM D27), attenuated as a result of defects in its ability to acquire iron. The patient died of cardiac arrest in the Emergency Department (ED) after six days of febrile symptomatology. Post-mortem results of the blood cultures collected in the ED were consistent with *Y. pestis*, later on characterized as the KIM D27 strain, the same strain the patient worked with in the laboratory. During the autopsy the patient was suspected to be affected by hereditary haemochromatosis, an iron overload disease, the diagnosis of which was then genetically confirmed. A proposed explanation for the unexpected fatal outcome was that the iron overload could have provided sufficient iron to the attenuated KIM D27 strain, thus enabling it to overcome its limitation and become virulent [161,162].

The characteristic of *Y. pestis* of being hidden to the immune system seems to be due to several factors. These include its capacity to target phagocytic cells (macrophages, dendritic cells, and neutrophils) and make them unresponsive by injecting the Yops immunosuppressive virulence factors, thus silencing both innate and adaptive immunity [163].

A further virulence factor with anti-phagocytic activity is the pH 6 antigen (pH 6 Ag; PsaA), which is physically expressed as fimbriae, in acidic media, at 37 °C [164]. Moreover, the structure of its lipopolysaccharide (LPS), which is normally hexa-acylated in the flea environment at 26 °C, becomes tetra-acylated after the transition to the mammalian environment at 37 °C. Tetra-acylated LPS results in a very weak Toll-like receptor (TLR)-4 stimulator, unable, therefore, to strongly activate the innate immune system, as the rough hexa-acylated lipid A with side chains 12–14 carbons in length of LPS maximally does [165,166]. It has been shown that the modified LPS, unable to strongly stimulate the TLR-4, seems to be the most relevant virulence factor of *Y. pestis* [165] (Figure 4).

Another characteristic of *Y. pestis* LPS is the absence of O-antigen [167], which facilitates invasion of dendritic cells expressing the Langerin receptor; the infected dendritic cells are relevant for infection dissemination, thus further contributing to bacterial virulence [168].

Although it has long been known that *Y. pestis* can easily survive inside macrophages [169], but it is generally killed inside neutrophils [155], it has been observed that the virulence plasmid pCD1 enables the bacteria to evade the host’s neutrophil response, which are rapidly recruited to the site of bacterial entry [170] thereby reducing neutrophil-mediated protection.

Moreover, through a pCD1-independent mechanism, *Y. pestis* may resist neutrophil phagocytosis, considering that ~70% of neutrophils 12 h after *Y. pestis* infection express the apoptotic marker phosphatidylserine on their surface, thus signaling that they are ready to be phagocytosed through efferocytosis by macrophages. Macrophages may, therefore, engulf, transport, and disseminate viable *Y. pestis*, while simultaneously limiting the production of pro-inflammatory cytokines, probably due to the increased secretion of the anti-inflammatory cytokine interleukin-1 receptor antagonist (IL-1RA) [168,171]. The Yop-mediated inhibition of phagocytosis, production of reactive oxygen intermediates, and macrophage activation also inhibits the release of pro-inflammatory cytokines, with a lack of inflammation characterizing the first days of disease after infection.

In addition, *Y. pestis* may also prevent complement activation, both at 26 °C and 37 °C, an activity mediated by the outer membrane protein Ail (attachment invasion locus), which works as adhesin/invasin [172]. More recently, these observations have been confirmed by also identifying the mechanism of action of Ail, which binds C4 binding protein, and, in the presence of factor I, may cleave C4b already bound to *Y. pestis*, thus blocking the important innate humoral defense mechanism of complement [173]. Natural killer (NK) cells are globally depleted (blood, spleen, and liver) as a consequence of YopM virulence factor [174]. NK cells are the main producers of interferon (IFN)-γ, whereas macrophages and endothelial cells produce tumor necrosis factor (TNF)-α and IL-8; the *Y. pestis* factors of virulence, such as Yops, inhibit the production of these pro-inflammatory cytokines, thus even reducing the level of inflammation in the environment which is favorable to the development of adaptive immunity. *Y. pestis* may markedly inhibit inflammation even through the YopM virulence factor, which specifically inhibits the pyrin-inflammasome complex [175], thus blocking the IL-1β release.

Based on this observation, it has been hypothesized that the appearance of the mutations in the *MEFV* gene, which underlie familial Mediterranean fever (FMF) in the eastern mediterranean region approximately 2600–1800 years ago, might have represented a *Y. pestis*-induced genetic adaptation to resist the serious infection [176]. The *MEFV* gene encodes the pyrin protein, which belongs to the cytosolic pattern recognition receptors (PRRs) and, upon activation, may assemble multiprotein complexes called inflammasomes, able to recruit and activate caspase 1, a pro-inflammatory protease [177] and a relevant antibacterial defense component of the innate immunity. The mutations of the *MEFV* gene found in the FMF allow to synthetize mutated pyrin which are resistant to the YopM-mediated inhibition of inflammasome activation, thus letting caspase 1 activation, IL-1β release, pyroptosis [178] and protection from *Y. pestis*. The permanence of *Y. pestis* inside the niche of phagolysosome in the macrophages at 37 °C allows bacteria to synthetize other virulence factors, such as F1, which creates a sort of capsule around the bacterial body, thus allowing it to acquire resistance to phagocytosis during the extracellular phase of the infection [166]. The relevance of Pla as virulence factor is evident for infection dissemination in bubonic plague [179]; moreover, it allows *Y. pestis* to replicate in the airways so rapidly as to induce fulminant pneumonic plague [180]. However, the high case-fatality rate of pneumonic plague is even due to the late arrival of neutrophils. This delay results from the anti-inflammatory action of YopJ, which blocks the release of chemotactic factors during the first 24 h. Conversely, at 72 h post-infection pro-inflammatory cytokines and chemokines increase significantly, even 10–100-fold as observed in a mouse model of pneumonic plague intranasally infected. This anti-inflammatory early phase is absent when an avirulent *Y. pestis* strain is used [181]. When analyzing the cellular and humoral immune response to Pla in humans immunized with live plague vaccine it could be demonstrated that the cellular response is mainly represented by Th1 and Th17 cells, and IL-17A, mainly produced by neutrophils, seems to represent a reliable indicator of induced immunity [182].

Adaptive T and B lymphocyte immunity has been demonstrated to be inhibited by YopH [183]. Such an immune system paralysis allows *Y. pestis* to multiply uncontrolled in the first 3–4 days after infection.

An animal model of bubonic plague in the rat has allowed to reconstruct the kinetics of progression of the infection by *Y. pestis*, which in the enlarged lymph node (bubo) logarithmically multiply in the absence of any innate immune response and induce apoptosis, so that the lymph node appears necrotic and hemorrhagic. After a few days, bacteria are extracellular and reach the bloodstream, thus disseminating the infection to secondary lymph nodes, spleen, liver and the whole body, with either a septicemic picture or lung localization, as secondary pneumonic plague [151].

### 7.2. Transmission and Clinical Forms

*Y. pestis* may be transmitted by the bite of an infected flea, or, more rarely, by the handling of an infected rodent or by the inhalation route, in case of pneumonic plague or use as a biological weapon (Figure 5).

Three main clinical forms of the disease are known, the most frequent of which (70–90%) is the bubonic plague, characterized by the sudden appearance of high fever, chills, myalgia, weakness, general malaise, nausea, headache, accompanied by quick, huge, and painful enlargement of draining lymph nodes (buboes) up to 1–15 cm in diameter after a brief incubation period of generally 2–3 days, but periods of 1 to 5 days are not infrequent [184,185].

The legs are particularly exposed to the flea bites; thus, the involvement of inguinal lymph nodes is the most frequent, followed by axillary and cervical lymph nodes [185]. If left untreated, the disease has a lethality of approximately 60% and/or in approximately 30% of cases may progress to a septicemic syndrome, due to bacterial entry into the bloodstream, with poor prognosis and higher lethality up to 90% [184].

Septicemic plague may be associated with disseminated intravascular coagulation, with necrosis of small vessels and gangrene of acral regions, from which the name Black Death, as the second pandemic was named [186]. However, it should be underlined that in a recent systematic review analyzing 1343 cases of bubonic plague described between 1902 and 2021, the general case-fatality rate was 15% [185].

Finally, the third and most severe form of the disease is the pneumonic plague, which may be observed in no more than 5% of cases, due to secondary localization in the lung or to primary infection acquired through the inhalation route, including release as a biological weapon. It is almost always fatal within three days if left untreated, whereas the survival rate may be between 25 and 50% if the right antibiotic treatment is started within 24 h from the beginning of the symptomatology [28,187].

In pneumonic plague, the human-to-human infection as a consequence of cough is possible, whereas no human-to-human contagion occurs in the bubonic and septicemic plague [159]. Consequently, patients with pneumonic plague should be isolated.

Although pneumonic plague is rather rare, in the first half of the last century large outbreaks of pneumonic plague have been observed in Manchuria, in 1910–1911 (involving as many as 60,000 people) and in 1920–1921, and in India, in the same period with 1400 deaths [186].

Even recently, in Madagascar in 2017, an outbreak of approximately 2400 plague cases, ~78% of whom were in pneumonic form, with 202 deaths was observed [100]. More rarely, even plague meningitis or plague, pharyngitis may occur. In the latter case cervical lymphadenopathy is associated [186].

## 8. *Yersinia pestis* as a Biological Weapon

Due to its high virulence and pathogenic potential, *Y. pestis* has long been considered an ideal candidate for biological warfare. Currently, considering the possible use of a multidrug-resistant strain, it is included among the Category A biological weapons [188].

Already in the mid-14th century, the Tatar army, during the siege of the Caffa fortress in Crimea, attempted to conquer the city by catapulting the corpses of plague victims over the city walls [188].

In World War II, the Japanese secret branch Unit 731, which specialized in the development of biological and chemical weapons, is reported to have released plague-infected fleas over densely populated areas in China, causing localized outbreaks of plague [186].

During the Cold War, both the United States and the Soviet Union actively engaged in offensive biological weapons programs, developing techniques to directly release aerosols of *Y. pestis* instead of unpredictable plague-infected fleas. This method enabled the direct induction of the more severe pneumonic plague, in contrast to the bubonic form typically caused by flea bites. However, only Soviet scientists successfully completed such offensive programs, obtaining large quantities of weaponized multidrug-resistant plague bacteria. In contrast, the US unilaterally terminated their offensive biological programs in 1969, without having produced significant quantities of weaponized plague bacteria [186].

The offensive potential of aerosolized weaponized *Y. pestis* was shown in a model elaborated by the WHO, in which the release of 50 kg of weaponized plague bacteria as an aerosol over a city of 5 million inhabitants was estimated to cause up to 150,000 cases of pneumonic plague and approximately 36,000 (24%) deaths.

The bacilli were estimated to remain viable in the air for 1 h for a distance of 10 km [189]. Clusters of pneumonic plague occurring in non-enzootic areas, without preceding rodent deaths, and in individuals lacking risk factors may suggest a deliberate release [186].

## 9. Diagnosis

According to the WHO, the diagnosis of plague may be defined as a suspected, probable, or confirmed case [190].

A *suspected case* is defined by the presence of symptoms compatible with plague and associated with compatible epidemiological data, such as exposure to plague infected animals or humans, and/or evidence of flea bites and/or residence in, or travel to, a known endemic area within 10 days prior to the onset of the disease.

A *probable case* has the characteristics of a suspected case with one of the following additional criteria: direct microscopy of a clinical sample, positive for Gram-negative coccobacilli that display bipolar staining with Wayson or Giemsa stain; positive F1 antigen in bubo aspirate, blood, or sputum; a single anti-F1 serology positive without evidence of previous *Y. pestis* infection or vaccination.

Finally, the *confirmed case* has the characteristics of a suspected case and at least one of the following criteria: (1) Isolation of *Y. pestis* from a clinical sample, which must have appropriate colony morphology and be identified as *Y. pestis* based on at least two of the following: (a) phage lysis at 20–25 °C; (b) biochemical profile; (c) F1 antigen detection; (2) Seroconversion or a 4-fold difference in anti-F1 antibody titer in paired serum samples drawn at least 2 weeks apart; (3) *Y. pestis* DNA detected by species-specific Polymerase Chain Reaction (PCR) on either clinical sample or culture according to standard practice [190].

The laboratory diagnostic gold standard is isolation and identification of *Y. pestis*, which should be performed at minimum in a Biosafety Level 3 laboratory, on standard culture media.

However, the use of a selective agar supplemented with CIN favors the isolation of the bacterium in polymicrobial samples such as sputum. After 2 or 3 days of incubation at 28 °C, the colonies formed are small, with a diameter of 1 to 2 mm, raised centers and a flat periphery. Gram staining reveals *Y. pestis* as Gram-negative small pleomorphic rods, while Giemsa or Wayson staining show them as bipolar coccobacilli [95]. The bacterial culture also allows the bacterial strain to be tested for antibiotic sensitivity.

However, culture of biological samples such as bubo aspirate, blood, or sputum, depending on the clinical form of the disease, is lengthy, while the speed of diagnosis is precious, particularly in pneumonic plague, so that appropriate antibiotic therapy can be started as soon as possible.

At any rate antibiotic therapy must be initiated on the first day of symptomatology, even though the diagnosis has not yet been confirmed. The cultures could be specifically lysed by *Y. pestis* phage at 22 to 25 °C. The bacteriophage φA1122, which is recommended by the Centers for Disease Control and Prevention (CDC), is widely used for diagnostic purposes thanks to its high level of specificity and its ability to lyse almost all-natural *Y. pestis* isolates. This phage infects and lyses *Y. pestis* cells, producing clear plaques on agar plates, which serves as a confirmatory test for the presence of the pathogen.

However, its diagnostic efficacy is limited in complex clinical matrices (e.g., blood), where serum factors reduce its lytic activity [191].

To overcome these limitations, φA1122 has been genetically engineered with the luciferase (*luxAB*) gene, yielding a bioluminescent detection system capable of identifying bacterial loads below 10^5^ colony-forming units (CFU)/mL in under 4 h [192]. This approach can be miniaturized for point-of-care devices, improving diagnostic access in resource-limited settings. Moreover, it has been refined by integrating phage lysis with quantitative real-time PCR (qPCR) to monitor the amplification of phage DNA during infection.

This method provides a rapid, highly sensitive, indirect way of detecting live *Y. pestis* cells within approximately four hours, with φA1122-based qPCR capable of detecting as few as one bacterial cell per microliter. This phage-qPCR assay bypasses the need for DNA extraction and can detect *Y. pestis* even in complex samples such as blood, representing a faster, more sensitive and specific detection than traditional phage lysis tests [193].

Another diagnostic phage, L-413C, is considered even more specific, while less sensitive, and is commonly employed in Europe and Central Asia. Using both phages in parallel enhances diagnostic accuracy. Additionally, other phages with more favorable biochemical properties have been selected, such as phage PST, which retains lytic activity even in the presence of human blood [191]. This makes it ideal for direct diagnostics from biological fluids, bypassing the need for pre-culturing. In 2024, the YPP 401 cocktail was developed, comprising four phages effective against 68 clinical and environmental *Y. pestis* strains with high genetic variability [188]. Although originally designed for post-exposure therapy, YPP 401 has demonstrated indirect diagnostic potential due to its broad genomic coverage. Combining phages with different receptor targets reduces the risk of false negatives caused by mutations in LPS genes or the PsaA protein, recently identified as the primary receptor for phage L-413C in a 2025 biophysical study [194].

In conclusion, phage-based diagnostics for *Y. pestis* represent a rapidly evolving technology that could complement or potentially replace conventional methods in emergency settings. F1 antigen and anti-F1 antibodies are analyzed with immunological methods, such as hemagglutination inhibition test, direct immunofluorescence, and enzyme-linked immunosorbent assay [95].

A quicker alternative to culture, which may not easily be available in some endemic countries, is the PCR targeting the genes of different virulence factors, including the F1 antigen gene (*caf1*), *pla* gene, or chromosomal fragments (for example, fragment 3a) [95,152]. However, recently *pla* gene and chromosomal fragment targets have been considered unreliable [95]. Immunological and PCR methods may be performed in a Biosafety Level 2 laboratory. Rapid tests have also been developed and used in endemic countries, based on dipsticks containing antibodies against F1 antigen.

The performance of these cheap, rapid (requiring only 15 min for results) and easy-to-use tests, thus particularly fit for endemic, low- and middle-income countries, has been recently evaluated by a Cochrane review, and the sensitivity and negative predictive value were found to be 100%, thus fit for excluding the pneumonic plague, whereas the specificity was 70% [195] (Table 3). The identification of *Y. pestis* is not easy and should be considered presumptive until confirmed by the reference laboratory, particularly in non-endemic countries.

An example of integrated One Health approach is an initiative to allow the serological surveillance in both humans and other mammalian hosts. The study of the serological status of humans, rodents, and domestic animals, such as dogs and cats, has been investigated using a newly developed rapid detection test for anti-F1 antibodies analysis by either hemagglutination (HA) or enzyme-linked immunosorbent assay (ELISA). *Staphylococcus* Protein A was used as a universal recognition reactant for humans, rodents and dogs. The results of this performance assessment study in Brazil showed that 16/43 humans (37%), 22/44 rodents (50%) and 18/54 dogs (33%) were positive, resulting in 100% of sensitivity and specificity [196].

## 10. Prophylaxis and Therapy

### 10.1. Vaccines

In 1897 Waldemar Mordecai Haffkine, a Russian biologist working at the Pasteur Institute in Paris, after having developed an effective, heat-killed, cholera vaccine, rapidly developed and produced a heat-killed anti-plague vaccine in India, which was successfully administered to 147 inmates at the Byculla House of Correction in Bombay, where a plague outbreak had exploded, whereas 172 were left untreated.

Among these untreated prisoners 12 cases and 6 deaths were registered, while only two cases and no deaths were observed in the treated ones (*p* = 0.03026, Yates corrected, two tails, χ^2^). Between 10 January and 6 May 1897, 11,362 people were vaccinated with the Haffkine’s vaccine in India; 45 of them fell ill with plague, in 12 cases fatal.

Although precise data of the attack rate in non-vaccinated people were not available, a rough estimate allowed us to calculate that the vaccinated subjects suffered 20-fold less than the non-inoculated ones [197]. However, the Haffkine’s vaccine was burdened by severe adverse effects and showed to be protective only against bubonic plague, not pneumonic one [158]. At the beginning of the twentieth century, following the successful pioneering observations of Kolle and Otto on the effectiveness of a small amount of live-attenuated *Y. pestis* microorganisms on experimental rodents [198], Strong reported that live-attenuated plague vaccines were also protective for human bubonic plague [199,200]. This vaccine has been shown to be effective when administered to millions of people in Indonesia, Madagascar, and Vietnam [201].

In contrast with the whole-cell inactivated vaccine, which is only protective against bubonic plague, the live-attenuated vaccine has been demonstrated to be effective against both pneumonic and bubonic plague [202]. The live-attenuated vaccine was obtained more than a century ago by isolating the bacterial strain from a girl with bubonic plague in Madagascar, whose name initials were EV, and by attenuating it in the 1920s through serial passages.

It and its derived strains, such as EV76, were administered to millions of people in the states of the former Soviet Union, in China, and Mongolia, showing a favorable safety profile, because no serious adverse events have been reported, a good immunogenicity and an appreciable protection, even in pneumonic plague [203].

The live-attenuated vaccine is more protective than the inactivated vaccine, even though the US military after World War II developed a formalin-killed vaccine which was successfully used for vaccinating the US military. In fact, it conferred near-complete protection during the Vietnam War [198,203] and was even licensed and commercialized in the USA with the name Plague Vaccine, USP [158].

However, both killed and live-attenuated vaccines are considered too reactogenic, thus they are not licensed in Europe and the formalin-killed plague vaccine was withdrawn even in the USA at the end of the last century.

More promising appeared to be the subunit vaccines, including the F1 anti-phagocytic and the T3SS component LcrV (low-calcium response virulence) factors together.

In 1998 US military medical researchers of the US Army Medical Research Institute for Infectious Diseases (USAMRIID) developed a recombinant vaccine, rF1-V, which was shown to be protective against both, bubonic and pneumonic plague, in experimental mice [204]. This vaccine, which is adsorbed onto the chemical adjuvant alhydrogel (alum) and is generally administered according to a prime-boost schedule, has also been demonstrated to be protective against pneumonic plague for rats, guinea pigs, and non-human primates [205]. Moreover, it has also been studied in humans in Phase 1 [206], Phase 2a [207,208], and Phase 2b (NCT05330624, ClinicalTrials.gov), in order to be approved by the Food and Drug Administration (FDA) [209].

Although this vaccine seemed a step forward compared to the killed whole cell and the live-attenuated vaccines as far as safety and protection are concerned, and in one study immunogenicity was found to be maintained up to one year [208], uncertainty still remained on the actual duration of the induced protective immune response [210].

Consequently, many researchers have tried to improve the general performance of the subunit vaccine, principally by working on the adjuvant component [211]. Carvalho et al. [212] developed a vaccine with the outer membrane vesicles (OMVs) from the gut commensal bacterium *Bacteroides thetaiotaomicron*, which was bioengineered to express *Y. pestis* F1 and V. This vaccine was intranasally and orally administered to non-human primates; it was safe, highly immunogenic, because it elicited robust specific serum IgG and mucosal IgA able to kill *Y. pestis*, as well as pro-inflammatory cytokines, such as IL-6, IL-1β, and IL-23, and mobilizing cytokines, such as monocyte chemoattractant protein 1 (MCP-1) and IL-8.

Thus, the choice to use an innocuous, mucosal microorganism has allowed to obtain a safe and effective vaccine, which can be conveniently administered via the nasal or oral route to be addressed to the mucosal site particularly exposed to *Y. pestis*, and the OMVs have shown to even possess an intrinsic adjuvant activity [212].

Wagner et al., instead, used an rF1-V vaccine linked to a double adjuvant. This system consisted of a polyanhydride nanoparticle system, able to work as adjuvant and delivery vehicles, and the stimulator of interferon genes agonists cyclic dinucleotides (CDNs). They could observe a rapid (<14 days) and long-lived protective immune response in mice vaccinated with a single dose of this vaccine and challenged with a lethal amount of intranasally administered *Y. pestis* both, at 14 and 218 days post-immunization [213]. The role of TLR-4 and the signal adaptor protein myeloid differentiation primary response (MyD)88, particularly this latter, is crucial for an optimal immune response to rF1-V subunit vaccine and for protection against an aerosol challenge of *Y. pestis* CO92 in mice [214].

Frey et al. in a phase 1 study could observe that the use of flagellin as adjuvant, in place of alum, allowed to significantly spare the amount of antigen used (6–10 μg of antigen versus 30 μg or even more), while maintaining a good safety and immunogenicity [215]. CpG 1810 has also been recently tested in a phase 2 study (NCT05506969, ClinicalTrials.gov). Moreover, combined vaccines, including in their composition *Y. pestis* proteins F1-V and anthrax Protective Antigen, with or without Lethal Factor, have been developed and found to be protective against both pneumonic plague and anthrax [216,217].

Adenovirus-vectored vaccines have also been developed and experimentally and successfully tested. In particular, the trivalent capsular antigen F1, the T3SS LcrV antigen, and the needle protein YscF containing adenovirus 5-vectored vaccine has been developed and tested on mice and non-human primates. The vaccine was intranasally administered (priming) and two weeks later the alum-adjuvanted recombinant proteins were intramuscularly administered (boost); the prime-boosted animals were challenged two weeks later with an aerosol of *Y. pestis* CO92 strain, resulting in 100% of protection [218].

The same group could demonstrate five years later that the same trivalent adenovirus 5-vectored vaccine intranasally administered either one- or two-fold to mice resulted in total protection against pneumonic plague and induced robust serum and mucosal antibody response as well as cell-mediated Th1-type specific immunity [219].

Very recently researchers from Israel developed a bivalent F1-LcrV mRNA IgG-Fc-conjugate included in the lipid nanoparticles (LNPs) as delivery system. This innovative vaccine induced robust antibody and specific cell-mediated immunity, thus providing full protection to intranasally infected mice of both inbred and outbred strains [220].

This same group had already developed a monovalent F1 mRNA-LNP (lipid nanoparticles) vaccine, conjugated to IgG-Fc, which was able to provide full protection in a mouse model of bubonic plague [221].

However, despite such a dynamic and productive scientific activity, no phase 3 study has up to now been initiated, thus the possibility of having available a safe and effective vaccine complying with the WHO recommendations [222], which may, therefore, have the requirements to be approved by FDA, still appears to be a distant prospect.

### 10.2. Passive Immunization

Shortly after the discovery of passive immunization by Emil von Behring and Shibasaburo Kitasato in 1890 against tetanus and diphtheria [223,224] and soon after his identification of the etiological agent of plague in 1894, Alexandre Yersin developed, with the help of his colleagues of the Pasteur Institute in Paris Émile Roux, Albert Calmette and Amédée Borrel, a specific antiserum [225]. It was successfully tested in 1896 on 23 Chinese patients in Hong Kong, with only two fatalities reported, corresponding to a mortality rate of 9% [226].

Subsequent studies did not confirm such a success, and in some instances, serotherapy proved ineffective. Serotherapy was gradually phased out during the 1940s, following the discovery of the antibacterial efficacy of sulfonamides, and later streptomycin in the late 1940s was discovered and considered more effective than passive immunization [226].

However, although a careful analysis of the results of the serotherapy cannot be made as the antisera were derived from different animal sources, the antibody titers and doses were unknown, and the studies lacked randomization, a retrospective analysis by Meyer et al. involving more than 20,000 patients from Asia, Africa and South America, suggested that mortality among serotherapy-treated patients was 35%, compared to 82% among untreated individuals [226,227]. More recently, the interest in passive immunization has raised again, even to identify, through the production of monoclonal antibodies, the protective epitopes within the virulence factor molecules, and the UK group in Porton Down could demonstrate that the protein tract between the amino-acids 135 and 275 of the LcrV molecule is a dominant protective epitope [228]. Monoclonal antibodies directed against LcrV or F1 antigen or both have been demonstrated to be protective against bubonic and pneumonic plague in mice [91]. Although no monoclonal antibody for passive immunization has been approved yet, very recently a Phase 1 randomized, double-blind, placebo-controlled, single ascending dose and dose expansion study of safety, tolerability and pharmacokinetics of a monoclonal antibody, JST-010, in healthy adults has been registered at ClinicalTrials.gov with the code NCT06943378.

### 10.3. Antibiotics

In the late 1930s, sulfonamides were tested in plague patients in East Africa and India [226]. In East Africa six patients were treated, but only three survived, specifically those treated earlier [229], whereas in India three cases were treated and all of them survived [230].

Following initial case report successes, in Madagascar in the early 1940s, 37 cases of bubonic plague were treated with sulfapyridine, nine of whom (24%) died. All the eight patients with pneumonic plague who had received the sulfonamide died [231]. However, three cases of pneumonic plague were successfully treated with sulfapyridine in 1947 in Madagascar [226,232]. Streptomycin, which was discovered in 1947, represented the real step forward, considering that the mortality had decreased from 70% to 35% with the passive immunization using specific antiserum, then to 20% with the sulfonamides, and reached its lowest point of approximately 7% with streptomycin, a threshold that has not been surpassed since [225]. Moreover, the streptomycin was successful even in patients in a moribund state [233], could induce a faster defervescence compared to the sulfonamides [234] and seemed to be more active than the sulfonamides against pneumonic plague [226]. Other antibiotics active against *Y. pestis* are chloramphenicol, which penetrates the central nervous system (CNS) effectively, thus it is the antibiotic of choice in case of plague meningitis, and tetracyclines. Both of them were first used in Madagascar in 1953 [235] in cases of pneumonic plague. All patients treated within 20 h from symptom onset exhibited a markedly favorable clinical response, whereas two patients treated at 24 and 48 h died. Chloramphenicol and tetracycline were as effective as streptomycin, not better than it [226].

In 1973 co-trimoxazole was successfully used in 12 patients with bubonic plague (three with septicemia) as the only drug, generally at the dose of 2 tablets (each tablet containing 80 mg of trimethoprim and 400 mg of sulfamethoxazole) twice a day, but even at higher doses, generally for 5–11 days, but extended to 15–17 days in the cases with septicemia. All the patients survived [236]. Although co-trimoxazole appears to be a relevant therapeutic resource in plague, particularly in case of infection with multidrug-resistant *Y. pestis* strains [237], the direct comparison with streptomycin showed that defervescence was faster in the latter [238].

Due to the nephrotoxicity and ototoxicity of streptomycin, and its teratogenic risk during pregnancy, gentamicin has been tested for efficacy as a possible substitute [227].

A retrospective analysis on 50 patients, who had been treated in the USA with streptomycin (14 patients) compared to gentamicin alone, gentamicin plus tetracycline, or tetracycline alone in the period 1985–1999, showed that streptomycin and gentamicin, alone or in combination with tetracycline, had a very similar efficacy thus indicating gentamicin as a possible substitute for streptomycin [239]. This conclusion has been reinforced by the results of a randomized controlled comparison between gentamicin and doxycycline in Tanzania in 2002, in which only 2 patients on gentamicin died (6%) [240].

Fluoroquinolones are the most recent antibiotics found to be active against *Y. pestis* [241]. They are considered safer than the aminoglycosides streptomycin and gentamicin, which are associated with nephrotoxicity and ototoxicity, and tetracyclines, which may induce photosensitivity, esophagitis and secondary fungal infection, considering that the possible side effect of fluoroquinolones is tendinopathy, thus the treatment is cost-effective in the light of the severity of the clinical picture [1].

According to the WHO, in case of pneumonic or septicemic plague the first-line antibiotic options for adults are fluoroquinolones (ciprofloxacin, at the dose of 400 mg every 8 h intravenously or 750 mg every 12 h via the oral route, or levofloxacin, 750 mg intravenously or via the oral route once a day, or moxifloxacin, at the dose of 400 mg intravenously or via the oral route once a day) and aminoglycosides (streptomycin 1 gr every 12 h intramuscularly or intravenously or gentamicin 5 mg/kg once a day intravenously or intramuscularly), whereas doxycycline should be considered an alternative option, at the initial loading dose of 200 mg, followed by 100 mg every 12 h intravenously or via the oral route [1] (Table 4).

In children the first-line options are the same as for adults, with the exception of moxifloxacin, which, being more reactogenic, is only suggested for children and adolescents as an alternative option; obviously, the doses of moxifloxacin and other antibiotics are reduced compared with the adults (Table 4).

In bubonic plague, the suggested first-line options are the same as for pneumonic or septicemic plague, at the same doses as for pneumonic or septicemic plague, except for doxycycline, which is reported among the first-line options, at the loading dose of 200 mg followed by 100 mg twice a day per oral route or intravenously in adults and children ≥45 kg (Table 5).

As alternative options, in addition to moxifloxacin for children and adolescents, tetracycline, at the dose of 500 mg (10 mg/Kg, maximum 500 mg/dose, in children >9 years) every 6 h via the oral route, chloramphenicol, at the dose of 12.5–25 mg/kg (maximum 1 g/dose), in adults and children >1 year, intravenously or via the oral route every 6 h, and sulfamethoxazole + trimethoprim, at the dose of 5 mg/kg (trimethoprim), in adults and children >2 months, intravenously or via the oral route every 8 h [1].

For all the forms of plague the antibiotic therapy should be administered for at least 10–14 days, or longer periods, if symptoms are still present.

In asymptomatic subjects who may have been exposed to *Y. pestis* in the previous 10 days a fluoroquinolone or a tetracycline per oral route may be administered for 7 days as a post-exposure prophylaxis [1].

Although antibiotic resistance in plague is rare, three plasmids have nonetheless been associated with resistance, two with monoresistance to streptomycin (pIP1203) [242] or doxycycline (pIP2180H) [243], and a third (pIP1202), which carries resistance genes to eight antibiotics, including the first-line options, such as streptomycin and tetracycline [237]. Consequently, the antibiotic associations are encouraged, given that antibiotic therapy must be initiated early, particularly in pneumonic and septicemic plague, before a confirmed diagnosis is established, thus it is necessary to expand the possibilities to use the right antimicrobial therapy.

Moreover, even the association between antimicrobial peptides and antibiotics is being explored to enhance antimicrobial efficacy and reduce antibiotic consumption, in an effort to mitigate side effects, such as nephrotoxicity and hearing loss caused by streptomycin [244].

A post-exposure combination of a single-dose live plague vaccine and a second-line antibiotic demonstrated efficacy in a murine model of pneumonic plague, demonstrating a synergistic effect between two individually suboptimal treatments [245].

Finally, given the limited number of randomized controlled trials on antibiotic use in plague, it should be underlined that three studies have been registered and completed at ClinicalTrials.gov, one managed by the US CDC (NCT00128466) Phase 2–3 on comparison between streptomycin and gentamicin, the second led by the CDC (NCT01243437) Phase 2 on the safety and efficacy of ciprofloxacin, and the third (NCT04110340) Phase 3 on the comparison between ciprofloxacin and streptomycin or gentamicin + ciprofloxacin, proposed by the University of Oxford.

### 10.4. Bacteriophages

Bacteriophages are viruses that infect and lyse bacteria with high specificity, often down to the strain level. They have been independently discovered by Frederik Twort (in 1915) and Felix d‘Herelle (in 1917) [246], and over a century ago, Félix d’Hérelle demonstrated that bacteriophages directly injected into the buboes of patients with bubonic plague could successfully cure the disease [247]. Bacteriophages are the most ubiquitous living organisms on Earth (~10^30^–10^31^) [248] and their highly selective activity is fundamentally distinct from that of antibiotics, thus allowing to consider their use even in association with antibiotics to increase the bactericidal effect [249].

Bacteriophages may be lytic or lysogenic. Lytic phages follow a replicative cycle which starts with the anchoring of the lytic phage to the bacterial surface specific receptor, then proceed with the injection of their genetic material into the bacterium, proceed with the viral replication and finally end with the bacterial lysis by resulting in release of progeny phages to target other bacterial cells [249]. Lysogenic bacteriophages, instead, integrate into the bacterial genome and they are inherited in the daughter bacterial cells. However, at a variable later time, as a consequence of some environmental stress, lysogenic bacteriophages excise from the bacterial genome and enter a lytic cycle [249]. Bacteriophages have been increasingly used in therapy since their discovery, frequently with success, up to the antibiotic development in the forties of the last century, when, particularly in the West, they were largely abandoned in favor of chemical antibiotics. In the Eastern countries, instead, including the former Soviet Union and Poland, the interest for bacteriophages was maintained, as even witnessed by the activity of the Giorgi Eliava Institute in Tbilisi, Georgia, which has been founded in 1923 with the help of Felix d’Herelle and is still active in the research and production of bacteriophages for diagnostic purposes, but primarily for therapeutic applications. The bacteriophage ՓA1122 is used for the diagnosis of *Y. pestis* by the CDC and the USAMRIID; moreover, there are several studies showing the in vitro activity of bacteriophages in the therapy of plague [250,251], whereas a lower number of studies may document the in vivo activity [246,252]. In addition, in one of these studies mice protected with bacteriophages alone and challenged with *Y. pestis* could only extend the survival time, but they did not survive unless a second-line antibiotic was added [246]. Thus, it may be hypothesized that in vivo interfering elements may intervene, including the possible resistance to bacteriophages, in analogy to what has been observed with antibiotics. Consequently, a phage cocktail targeting distinct bacterial surface receptors, may be a means for reducing the possibility of emergence of phage resistance [159,253]. Indeed, a four-phage cocktail, XPP-401, given intranasally or intraperitoneally 18 h after challenge with *Y. pestis* for intranasal route to Brown Norway rats showed ~88% of protection [188]. This indicates that a cocktail of bacteriophages using different bacterial surface molecules to anchor its target may be a productive way to face the relevant topic of the treatment of plague, particularly in the light of either naturally [237] or deliberately [254] occurring antibiotic resistance, in case of use of *Y. pestis* as a biological weapon. Moreover, bacteriophages may even be used in combination with antibiotics, to reciprocally potentiate the therapeutic effect [246].

## 11. Conclusions

Plague has historically been one of the most devastating diseases affecting humanity and continues to pose a serious threat, particularly due to concerns about its potential use as a biological weapon, especially in the form of multidrug-resistant strains [155,189]. The studies of paleogenomics have allowed us to trace the phylogenesis of *Y. pestis*, which, through gene acquisition or silencing, evolved from a gastrointestinally tropic microorganism into a vector-borne pathogen [130]. *Y. pestis* is rich in virulence factors able to transiently, but fully, block the host immune system, leading rapidly to host death [157]. *Y. pestis* is strictly interwoven with a very large series of animal susceptible hosts, particularly rodents [12], and vectors, particularly fleas, rendering its eradication extremely difficult, if not impossible. Moreover, a safe and effective human vaccine is still unavailable, underscoring the critical importance of rapid diagnosis and immediate initiation of antibiotic therapy, for reducing the high lethality [195]. The integration of the One Health approach, as a systems-thinking model integrating human, animal, and environmental health together with advances in vaccines, monoclonal antibodies, and bacteriophage therapy, offers tangible promise to reduce the threat of this terrible infection.

## Figures and Tables

**Figure 1 biomedicines-13-02555-f001:**
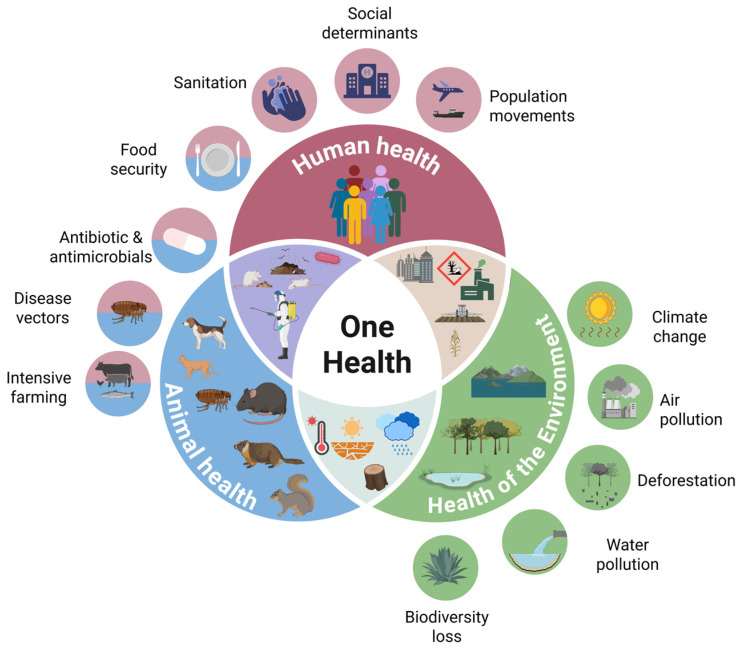
The One Health approach, which considers human and animal health together with the health of the environment in an integrated way. Created in BioRender. Di Spirito, M. (2025) https://BioRender.com/1bdq802, accessed on 2 July 2025.

**Figure 2 biomedicines-13-02555-f002:**
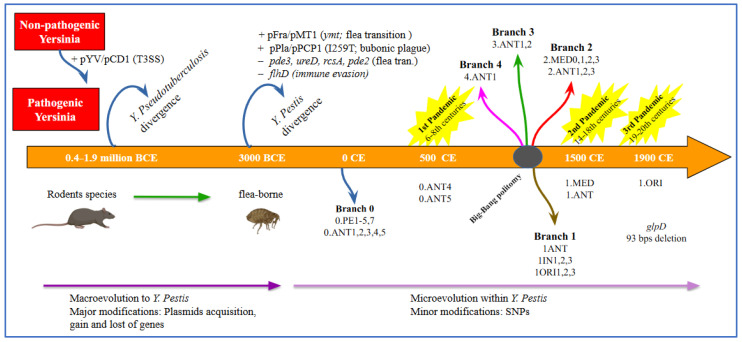
Schematic timeline of *Yersinia pestis* evolution. BCE = before the current era, PE = Pestoides, ANT = Antiqua, MED = Medievalis, IN = Intermediate, ORI = Orientalis. Partly created with Biorender.com.

**Figure 3 biomedicines-13-02555-f003:**
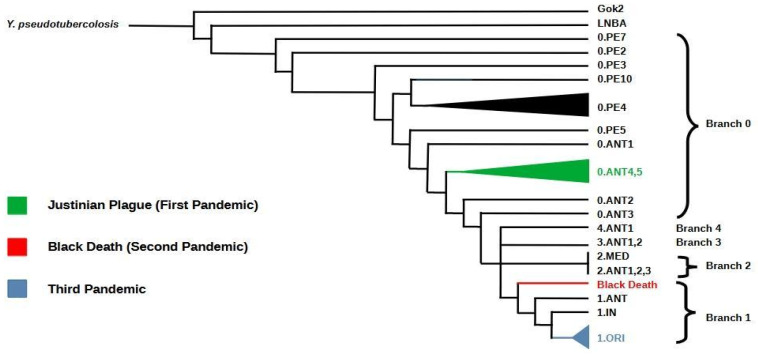
Simplified phylogenetic tree of *Y. pestis*, showing the main branches (0–4), their association with the Biovars, and the linkage with the three historical pandemics (Justinian Plague, Black Death, and Third Pandemic). The tree topology is adapted and simplified from Demeure et al. [111], with stylistic modifications for consistency with the present review.

**Figure 4 biomedicines-13-02555-f004:**
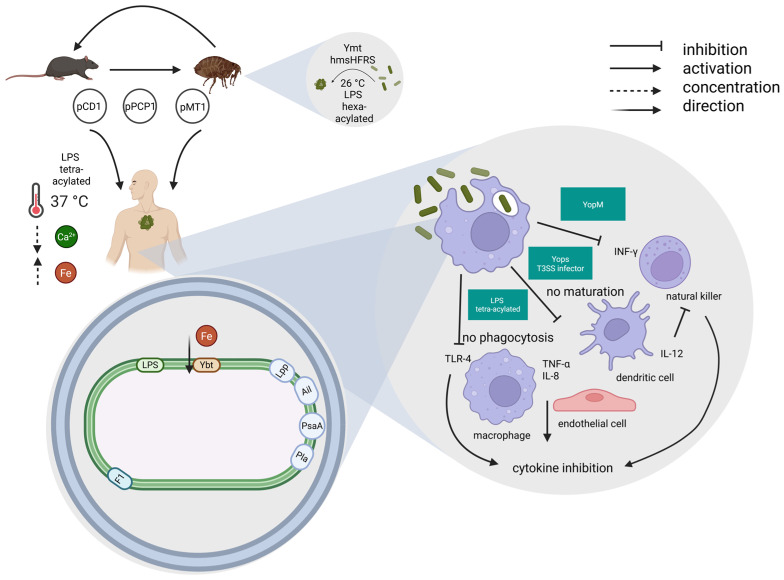
The main virulence factors of the *Y. pestis* and their interaction with the immune system are reported. The LPS is in its natural configuration inside the flea at 26 °C; however, in the infected human, at 37 °C, it becomes tetra-acylated, unable to strongly activate, through the TLR-4, the macrophage, with consequent block of phagocytosis and cytokine release. Even the Yops, products of the T3SS, contribute to the anti-phagocytic activity. LPS = lipopolysaccharide; TLR = Toll-like receptor; Yops = *Yersinia* outer-membrane proteins; T3SS = Type 3 secretion system. Created in BioRender. Di Spirito, M. (2025) https://BioRender.com/sdnyhv4, accessed on 2 July 2025.

**Figure 5 biomedicines-13-02555-f005:**
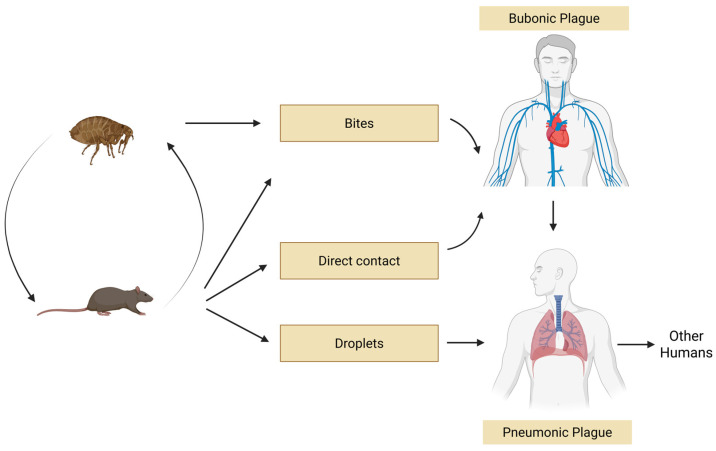
The transmission of the *Y. pestis* is reported. The most frequent way of transmission is through the bite of infected fleas, but even through the handling of infected animal hosts; these types of infection generally cause bubonic plague. In a minority of cases the infection may happen through air droplets, which cause pneumonic plague. In this case interhuman transmission is possible. Created in BioRender. Di Spirito, M. (2025) https://BioRender.com/swtkwqx, accessed on 2 July 2025.

**Table 1 biomedicines-13-02555-t001:** Key animal groups affected by *Yersinia pestis*.

Animal Group	Role in Plague Ecology
Rodents	Primary reservoirs; amplifying hosts during epizootics
Lagomorphs	Susceptible spillover hosts
Insectivores	Susceptible spillover hosts
Carnivores	Susceptible hosts; sentinels for plague activity
Other Mammals	Documented infections; potential transmission routes

**Table 2 biomedicines-13-02555-t002:** Plague pandemics and major epidemics.

Pandemic	Period	Affected Area	Estimated Deaths	Reference
First Plague Pandemic—Plague of Justinian	6th-8th century	Eastern Roman Empire, Eastern and Western Mediterranean and Western Europe	25–50 million	[144]
Second Plague Pandemic—Black Death	mid 14th–mid 18th century	Europe, Asia, North Africa	~30 million	[144]
Plague of Milan 1629–1631	1629–1631	Milan, Italy	~60,000	[139]
Great Plague of London	1665–1666	London, England	100,000–200,000	[141]
Plague of Marseille	1720–1722	Marseille, France	~100,000	[142]
Third Plague Pandemic	late 19th–mid 20th century	Global (originated in China, spread to India, Asia, Africa, and the Americas)	Over 26 million	[144]
Plague of Hong Kong	1894–1895	Hong Kong, China	~60,000	[144]

**Table 3 biomedicines-13-02555-t003:** Diagnostic techniques for plague.

Diagnostic Technique	Biosafety Level	Turnaround Time	Sensitivity/Specificity	Notes
Bacterial culture	BSL-3	2–3 days	High (gold standard)	Requires CIN selective medium; allows antibiotic susceptibility testing
Gram, Giemsa or Wayson staining	BSL-2	Rapid	Moderate	Detects pleomorphic Gram-negative/bipolar rods
F1 antigen detection (ELISA, immunofluorescence, hemagglutination)	BSL-2	Rapid	Good	Used on blood, buboes, sputum
PCR (caf1, pla, 3a)	BSL-2	Rapid	High, but some targets are no longer reliable	Rapid alternative to culture
Phage lysis test (φA1122, L-413C)	BSL-2/3	hours	φA1122: high; L-413C: more specific but less sensitive.	Also used with qPCR or modified with *luxAB*
Phage-qPCR (φA1122 + qPCR)	BSL-2	4 h	Very high	Detects 1 cell/μL without DNA extraction
Rapid immunochromatographic test (anti-F1 dipstick)	BSL-2	15 min	Sensitivity 100%, Specificity 70%	Suitable for endemic, resource-limited countries
Seroconversion or 4-fold change in anti-F1	BSL-2	≥2 weeks	High	Requires two serum samples at least 2 weeks apart
luxAB-modified phage (bioluminescence)	BSL-2	<4 h	High (under 10^5^ CFU/mL)	Can be miniaturized for point-of-care
Phage PST (active in blood)	BSL-2	hours	High in complex matrices	Does not require pre-culture; active in biological fluids
Phage cocktail YPP 401	BSL-2	hours	High (broad genomic coverage)	Indirect, diagnostic potential due to broad genomic coverage

CIN = cefsulodin–irgasan–novobiocin; ELISA = enzyme-linked immunosorbent assay; F1 = fraction 1 antigen; DNA = deoxyribonucleic acid; CFU = colony-forming units.

**Table 4 biomedicines-13-02555-t004:** First-line antibiotic treatment for pneumonic or septicemic plague (slightly modified from ref. [1]).

Antibiotics	Recipients	Dose Amount	Doses No.	Route of Administration
Ciprofloxacin	Adults	400 mg	3	Intravenous/oral
		750 mg	2	
	Children	10 mg/kg (max. 400 mg/dose)	2–3	Intravenous/oral
		15 mg/kg (max. 750 mg/dose)	2	
Levofloxacin	Adults	750 mg	1	intravenous/oral
	Children ≥ 6 months	<50 kg: 8 mg/kg (max. 250 mg/dose)	2	intravenous/oral
		≥50 kg: 500–750 mg	1	intravenous/oral
Moxifloxacin	Adults	400 mg	1	intravenous/oral
Gentamicin	Adults	5 mg/kg	1	intravenous/intramuscular
	Children	4.5–7.5 mg/kg	1	intravenous/intramuscular
Streptomycin	Adults	1 g	2	intravenous/intramuscular
	Children	15 mg/kg (max. 1 g/dose)	2	intravenous/intramuscular

**Table 5 biomedicines-13-02555-t005:** First-line antibiotic treatment for bubonic plague (slightly modified from ref. [1]).

Antibiotics	Recipients	Dose Amount	Doses No.	Route of Administration
Ciprofloxacin	Adults	400 mg	3	Intravenous/oral
		750 mg	2	
	Children	10 mg/kg (max. 400 mg/dose)	2–3	Intravenous/oral
		15 mg/kg (max. 750 mg/dose)	2	
Levofloxacin	Adults	750 mg	1	intravenous/oral
	Children ≥ 6 months	<50 kg: 8 mg/kg (max. 250 mg/dose)	2	intravenous/oral
		≥50 kg: 500–750 mg	1	intravenous/oral
Moxifloxacin	Adults	400 mg	1	intravenous/oral
Doxycycline	Adults and Children ≥ 45 kg	200 mg loading dose followed by 100 mg	2	intravenous/oral
	Children < 45 kg	4.4 mg/kg (max. 200 mg) loading dose followed by 2.2 mg/kg (max. 100 mg)	2	intravenous/oral
Gentamicin	Adults	5 mg/kg	1	intravenous/intramuscular
	Children	4.5–7.5 mg/kg	1	intravenous/intramuscular
Streptomycin	Adults	1 g	2	intravenous/intramuscular
	Children	15 mg/kg (max. 1 g/dose)	2	intravenous/intramuscular

## Data Availability

No new data were created or analyzed in this study. Data sharing is not applicable to this article.

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
