# Peer review of "Plagued by the Past, Pressed by the Present: A One Health Perspective on *Yersinia pestis"

_biomedicines, 2025, doi:10.3390/biomedicines13102555_

Round 1
Reviewer 1 Report
Comments and Suggestions for Authors
This review covers a broad array of topics on Yersinia pestis, including its ecology, infectious life cycle, natural history, pathogenesis, diagnostics, vaccine development, and antibiotic treatment. It offers a nice overview of the field, incorporating recent findings from both basic and clinical research, such as whole-genome phylogeny studies and vaccine clinical trials. The manuscript serves as an excellent introduction for readers outside the field and a valuable reference for researchers in the field. I also appreciated the authors’ emphasis on the importance of the “One Health” approach.
Below are some of my suggestions for improved readability:
- Some sentences are long and hard to follow (e.g., lines 1111–1118, 1125–1130, 1251–1257, and 1578–1584). Breaking them into shorter, clearer sentences would help.
- Lines 1236–1242: More background is needed for the discussion on MEFV gene mutations and their potential link to the resistance to pestis infection. It would be helpful to explain what this gene encodes and how the mutations affect host-pathogen interactions to clarify the significance of this example.
- Lines 1126-1152: In the section on the fatal laboratory-acquired infection in a researcher with hemochromatosis, a brief explanation of the KIM27 strain and why pgm- strains are considered avirulent in normal situation would make the discussion more informative. The authors should focus on the implications of the incident, rather than the detailed account of the event but.
- Reference #7 does not seem to discuss iSRS. Please review the reference list to ensure all citations correspond accurately to the sources cited in the text.
A few more optional suggestions on the organization of the review:
- There is some repetition, especially in Section 2. For example, the importance of the “One Health” approach is discussed multiple times. Additionally, the complex relationships between plague transmission and various environmental and socioeconomic factors are discussed repeatedly. Numerous examples are provided for each factor, but they only add limited value to the discussion. Reorganizing or consolidating some subsections could improve the flow and reduce redundancy.
- The topics in Sections 4–8 feel somewhat disjointed. It _might_ be clearer to group all content related to human disease (e.g., Sections 4, 6.2, 7.2, and 8) together, and follow with sections focused on microbiological and molecular aspects of the pathogen. (Though I am not too confident.)
- For topics not covered in detail (e.g., the Type III secretion system and Yop effectors), the authors could direct readers to other relevant reviews within the text.
Author Response
We wish to thank the reviewer for dedicating time to reviewing our work and for the valuable comments and suggestions. We have carefully considered and incorporated the requested changes, enhancing the clarity of our paper.
In the revised manuscript, the text that has been added or modified is written in green whereas the deleted lines are indicated by crossout.
- Comment 1: Some sentences are long and hard to follow (e.g., lines 1111–1118, 1125–1130, 1251–1257, and 1578–1584). Breaking them into shorter, clearer sentences would help.
Response 1: According to your comment we modified the following sentences to make them shorter and clearer, as shown below:
Lines 1111-1118 in revised version lines 1080-1086: “The other two plasmids are pPCP1 and pMT1. pPCP1 (pesticin, coagulase, plasminogen activator), also designated pPst, pPla, or pYP, of ⁓9.5 Kilobases of DNA, encodes for a plasminogen activator (Pla), that also works as adhesin, and bacteriocin pesticin (Pst). pMT1 (murine toxin), even designated pFra/Tox, pFra, pTox, or pYT, of ⁓100-110 Kilobases of DNA, encodes for Yersinia murine toxin (Ymt) and capsular protein fraction 1 (CaF1), this last one equipped with adhesin and antiphagocytic activity. “
Lines 1125-1130 in revised version lines 1093-1098: “ In addition to Ymt, the four proteins encoded by the hemin storage (hms)HFRS locus also act at 26°C to favor the permanence of Y. pestis in the flea midgut, being involved in the production of biofilm matrix exopolysaccharide. This strengthens the resistance of Y. pestis, which is allowed to multiply and form an aggregate that fills the proventriculus, eventually blocking the flea midgut.”
Lines 1251-1257 in revised version lines 1134-1138: “The characteristic of Y. pestis of being hidden to the immune system seems to be due to several factors. These include its capacity to target phagocytic cells (macrophages, dendritic cells, and neutrophils) and make them unresponsive by injecting the Yops immunosuppressive virulence factors, thus silencing both innate and adaptive immunity.”
Lines 1578-1584 in revised version lines 1545-1552: “Wagner et al., instead, used an rF1-V vaccine linked to a double adjuvant. This system consisted of a polyanhydride nanoparticle system, able to work as adjuvant and delivery vehicles, and the stimulator of interferon genes agonists cyclic dinucleotides (CDNs). They could observe a rapid (<14 days) and long-lived protective immune response in mice vaccinated with a single dose of this vaccine and challenged with a lethal amount of intranasally administered Y. pestis both, at 14 and 218 days post-immunization.”
- Comment 2: Lines 1236–1242: More background is needed for the discussion on MEFV gene mutations and their potential link to the resistance to pestis infection. It would be helpful to explain what this gene encodes and how the mutations affect host-pathogen interactions to clarify the significance of this example.
Response 2: According to your suggestions, we extended the discussion about the MEFV gene, reported on lines 1236–1242 in revised version lines 1211-1219.
- Comment 3: Lines 1126-1152: In the section on the fatal laboratory-acquired infection in a researcher with hemochromatosis, a brief explanation of the KIM27 strain and why pgm- strains are considered avirulent in normal situation would make the discussion more informative. The authors should focus on the implications of the incident, rather than the detailed account of the event but.
Response 3: According to your suggestions, we revised the text including a brief explanation of KIM D27 avirulence and to detail the potential role of iron overload in restoring virulence, in revised version lines 1119-1138.
- Comment 4: Reference #7 does not seem to discuss iSRS. Please review the reference list to ensure all citations correspond accurately to the sources cited in the text.
Response 4: We apologize. The correct reference is the following Zinsstag, J.; Crump, L.; Schelling, E.; Hattendorf, J.; Maidane, Y.O.; Ali, K.O.; Muhummed, A.; Umer, A.A.; Aliyi, F.; Nooh, F.; et al. Climate change and One Health. FEMS microbiology letters 2018 Jun 1;365(11):fny085. doi: 10.1093/femsle/fny085, currently indicated as number 6. All the other references have been checked according to the reviewer’s suggestion.
- Comment 5: There is some repetition, especially in Section 2. For example, the importance of the “One Health” approach is discussed multiple times. Additionally, the complex relationships between plague transmission and various environmental and socioeconomic factors are discussed repeatedly. Numerous examples are provided for each factor, but they only add limited value to the discussion. Reorganizing or consolidating some subsections could improve the flow and reduce redundancy.
Response 5: We thank the reviewer for pointing out redundancies. Actually the authors revised section 2 in order to improve the flow and reduce redundancy.
- Comment 6: The topics in Sections 4–8 feel somewhat disjointed. It _might_ be clearer to group all content related to human disease (e.g., Sections 4, 6.2, 7.2, and 8) together, and follow with sections focused on microbiological and molecular aspects of the pathogen. (Though I am not too confident.)
Response 6: We considered grouping all content related to human diseases together, and recognize that current formulation may generate some degree of ambiguity. However, we believe that this represents the best compromise to accurately reflect the One Health concept. In fact, if we take out all the paragraphs on humans, the human component of the One Health approach is lacking, only remaining the animal and environmental components.
- Comment 7: For topics not covered in detail (e.g., the Type III secretion system and Yop effectors), the authors could direct readers to other relevant reviews within the text.
Response 7: We addressed the comments by indicating the references 91 and 103 at lines 839-840
Along with the cover letter, we submitted:
The final revised manuscript: biomedicines-3767536_resubmission_Round1.docx, containing the fully final revised text formatted according to journal guidelines.
The annotated version: Biomedicines_3767536_tracking.pdf, containing the original text and all modifications: the original text is shown in black, the newly added or modified text is highlighted in green and the deleted text is shown as strikethrough. This annotated version is provided solely to facilitate the editorial and peer review process and is not intended for publication. Please see attachment.
Best Regards

Reviewer 2 Report
Comments and Suggestions for Authors
Authors have written a comprehensive, well-organized review that presents a timely synthesis of the historical, ecological, microbiological, and public health dimensions of Yersinia pestis through the lens of the One Health approach. The manuscript demonstrates strong interdisciplinary scholarship and effectively contextualizes plague as a persistent and evolving threat in both endemic and non-endemic regions. It integrates molecular epidemiology, environmental risk factors, surveillance strategies, and socio-economic drivers into a cohesive narrative.
The manuscript is suitable for publication pending minor revisions as outlined below.
Comments:
- Add References in the Introduction and Discussion: While the manuscript is extensively cited throughout, the Introduction and Conclusion/Future Directions sections would benefit from more recent and high-impact references (e.g., WHO or CDC updates, recent genomic studies from 2023–2024).
- Improve Figure Resolution: Figures 1 and 2 appear to be of suboptimal resolution and may not meet publication standards. Please provide high-resolution versions suitable for typesetting.
- Add Error Bars to Figure 4 (Left Panel): Ensure that the left-hand side of Figure 4 includes error bars where applicable. If these are unavailable or not applicable, a brief explanation should be added to the figure legend.
- Terminology Clarification: The terms 'enzootic' and 'epizootic' are used frequently—consider providing a concise definition in a figure or glossary.
- Consistency in Citations: Ensure uniform citation style, particularly for references cited as "[58]" vs. "[58,97,112]". Clustered citations should follow journal formatting guidelines.
- Typographical Clean-up: Occasional formatting artifacts such as "mailto:" links and line breaks may require removal in the final copy.
Author Response
We sincerely thank you for your kind and encouraging considerations.
According to your comment we modified the following sentences to:
- Comment 1: Add References in the Introduction and Discussion: While the manuscript is extensively cited throughout, the Introduction and Conclusion/Future Directions sections would benefit from more recent and high-impact references (e.g., WHO or CDC updates, recent genomic studies from 2023–2024).
Response 1: According to your suggestions, we have added further reference to the manuscript, for example actual ref. 139. The reference 7 has been renamed as 1 and moved in the introduction.
- Comment 2: Improve Figure Resolution: Figures 1 and 2 appear to be of suboptimal resolution and may not meet publication standards. Please provide high-resolution versions suitable for typesetting.
Response 2: We thank you for this observation. Figures 1 and 2 were created using BioRender at the highest resolution to ensure they meet publication requirements standards.
- Comment 3: Add Error Bars to Figure 4 (Left Panel): Ensure that the left-hand side of Figure 4 includes error bars where applicable. If these are unavailable or not applicable, a brief explanation should be added to the figure legend.
Response 3: We thank you for this comment. However, we are not fully certain about the request regarding the addition of error bars to the left panel of Figure 4 (currently Fig 5), because error bars are not applicable in this figure.
Comment 4: Terminology Clarification: The terms 'enzootic' and 'epizootic' are used frequently—consider providing a concise definition in a figure or glossary.
Response 4: We thank you for this comment. The definitions of 'enzootic' and 'epizootic' are now provided within the relevant paragraph, and we have removed any instances of these terms appearing before explanation.
- Comment 5: Consistency in Citations: Ensure uniform citation style, particularly for references cited as "[58]" vs. "[58,97,112]". Clustered citations should follow journal formatting guidelines.
Response 5: We thank you for this suggestion. We have ensured that all citations are now formatted consistently according to the journal’s guidelines.
- Comment 6: Typographical Clean-up: Occasional formatting artifacts such as "mailto:" links and line breaks may require removal in the final copy.
Response 6: We thank you for this observation. We have carefully checked the manuscript but we did not find any remaining formatting artifacts in our revised manuscript.
Along with the cover letter, we submitted:
The final revised manuscript: biomedicines-3767536_resubmission_Round1.docx, containing the fully final revised text formatted according to journal guidelines.
The annotated version: Biomedicines_3767536_tracking.pdf, containing the original text and all modifications: the original text is shown in black, the newly added or modified text is highlighted in green and the deleted text is shown as strikethrough. This annotated version is provided solely to facilitate the editorial and peer review process and is not intended for publication. Please see the attachment.
Best Regards

Reviewer 3 Report
Comments and Suggestions for Authors
This review titled “Plagued by the Past, Pressed by the Present: A One Health Perspective on Yersinia pestis” aims, according to the authors, to provide a comprehensive and updated overview of Y. pestis, with particular focus on the One Health model and recent advancements in whole genome-based typing, diagnostic technologies, and countermeasure strategies against its potential use as a biothreat agent. Overall, the review covers the most important topics on the subject and provides contemporary information. However, it lacks sharp focus in some areas and strays from its title and stated aims. Over their 50 pages of text the authors cover one health extensively in the first 5 pages, mention the topic again on page 18, and then not again until the last sentence on page 50. Given the emphasis on one health in the title, it is important to tie this subject to each category as much as possible. In addition, I have the following recommendations:
- Obtain assistance with grammar from a scientific editor. In addition to incorrect grammar usage that obscures the authors’ intended messages, extra words are frequently used.
- The term “plague” should only be used to describe a disease and Y. pestis should be used to describe the etiological organism. For example, line 846 “global phylogeny of plague”.
- The introduction to section 2 (lines 102-164) adds unnecessarily to the length of the paper with a discussion of one health that does not directly address the plague and should be shortened or omitted.
- Lines 632-646 are unrelated to the topic of epidemiological surveillance. All content in this section should be related to this subject.
- From line 975, the extensive description of the branches of a phylogenetic tree should be accompanied by the actual tree.
- Yops is defined on line 888 as outer-membrane proteins, on line 893 as yersinia outer proteins, and line 1108 as yersinia outer proteins. Abbreviations should be consistent.
- Line 1136, the description of the fatal case of plague in a laboratory worker should be referenced in the paragraph in which it is first described.
- Line 1196, The statement about the absence of Y. pestis LPS O-antigen is oversimplified. Please see Biomolecules. 2021 Sep 26;11(10):1410. doi: 3390/biom11101410
- Line 1222, Observations, not “Data”, are not being presented.
- Line 1226, What is meant by “NK cells are markedly reduced”? Reduced in number? Reduced activity?
- Line 1379, capitalize “Gram”.
- Line 1566, The genus and species should be in italics.
- Line 1696, Please specify the amount of co-trimoxazole per tablet.
- Line 1833, I assume “(~1030-1031 phage particles)” should be in exponential notation.
Comments on the Quality of English Language
The grammar requires extensive editing.
Author Response
We wish to thank the reviewer for dedicating time to reviewing our work and for the valuable comments and suggestions. We have carefully considered and incorporated the requested changes, enhancing the clarity of our paper. We have carefully revised the manuscript to ensure that the One Health perspective is consistently integrated throughout the text. References to One Health have been reinforced to better align the content with the title and stated aims of the review.
According to your comment we modified the following sentences to:
- Comment 1: Obtain assistance with grammar from a scientific editor. In addition to incorrect grammar usage that obscures the authors’ intended messages, extra words are frequently used.
Response 1: We apologize for the grammar errors, all the manuscript has been carefully revised by an English native speaker.
- Comment 2: The term “plague” should only be used to describe a disease and Y. pestis should be used to describe the etiological organism. For example, line 846 “global phylogeny of plague”.
Response 2: We have consistently replaced the term “plague” with “Y. pestis” where an etiological reference to the microorganism was required.
- Comment 3: The introduction to section 2 (lines 102-164) adds unnecessarily to the length of the paper with a discussion of one health that does not directly address the plague and should be shortened or omitted.
Response 3: Thank you for the comment and the suggestion. This part has been markedly reduced.
- Comment 4: Lines 632-646 are unrelated to the topic of epidemiological surveillance. All content in this section should be related to this subject.
Response 4: Following your suggestion, we have revised the paragraph title to better reflect its content by also including the previously omitted topic.
- Comment 5: From line 975, the extensive description of the branches of a phylogenetic tree should be accompanied by the actual tree.
Response 5: We added a simplified phylogenetic tree, according to the one published by Demeure et al. (Yersinia pestis and plague: an updated view on evolution, virulence determinants, immune subversion, vaccination and diagnostics, REF [115]), to support the description of the major biovars and their evolutionary relationships.
- Comment 6: Yops is defined on line 888 as outer-membrane proteins, on line 893 as yersinia outer proteins, and line 1108 as yersinia outer proteins. Abbreviations should be consistent.
Response 6: Thank you for your comment. We have consistently corrected the definitions.
- Comment 7: Line 1136, the description of the fatal case of plague in a laboratory worker should be referenced in the paragraph in which it is first described.
Response 7: The reference [165] has been moved to the first sentence introducing the fatal case.
- Comment 8: Line 1196, The statement about the absence of Y. pestis LPS O-antigen is oversimplified. Please see Biomolecules. 2021 Sep 26;11(10):1410. doi: 3390/biom11101410
Response 8: Thank you for your comment. We added the suggested reference [171] in the test.
- Comment 9: Line 1222, Observations, not “Data”, are not being presented.
Response 9: As you suggested, we have replaced the term "observations" with "data."
- Comment 10: Line 1226, What is meant by “NK cells are markedly reduced”? Reduced in number? Reduced activity?
Response 10: We have revised the sentence using the terminology of the original study. “reduced” is replaced with “depleted,” which more accurately refers to the observed decrease in NK cell numbers
- Comment 11: Line 1379, capitalize “Gram”.
Response 11: Thank you for your observation. The term "Gram" has been capitalized accordingly.
- Comment 12: Line 1566, The genus and species should be in italics.
Response 12: As you suggested, we have written genus and species in italics.
- Comment 13: Line 1696, Please specify the amount of co-trimoxazole per tablet.
Response 13:The text was revised to specify the amount of co-trimoxazole per tablet “...generally at the dose of 2 tablets (each tablet containing 80 mg of trimethoprim and 400 mg of sulphamethoxazole) twice a day…” at line 1661-1662
- Comment 14: Line 1833, I assume “(~1030-1031 phage particles)” should be in exponential notation.
Response 14: You’re right; we wrote the exponential notation correctly, line 1764.
Along with the cover letter, we submitted:
The final revised manuscript: biomedicines-3767536_resubmission_Round1.docx, containing the fully final revised text formatted according to journal guidelines.
The annotated version: Biomedicines_3767536_tracking.docx, containing the original text and all modifications: the original text is shown in black, the newly added or modified text is highlighted in green and the deleted text is shown as strikethrough. This annotated version is provided solely to facilitate the editorial and peer review process and is not intended for publication. please see the attachment.
Best Regards

Reviewer 4 Report
Comments and Suggestions for Authors
This review provides a systematic analysis of plague challenges across past, present, and future contexts through an integrative multidisciplinary lens encompassing microbiology, clinical medicine, epidemiology, ecology, and historical developments. A key focus is placed on the "One Health" framework, which emerges as the most effective paradigm for Yersinia pestis eradication by bridging human, animal, and environmental health sectors. The manuscript demonstrates exceptional organizational structure with clear thematic progression, a comprehensive coverage of critical aspects in plague research, and a strong scholarly foundation through extensive literature support. To prepare this manuscript for publication, the following major issues should be addressed:
- The current emphasis on the One Health approach in the title and introduction is not sufficiently reflected in the article's content structure. While section 2 addresses this framework, other substantial portions (virulence factors, Yersinia pestis evolution, clinical features, and treatment/prevention) remain disconnected from this central theme. To strengthen coherence, the authors should either revise the title and introductory statements to better match the actual content distribution, or significantly expand One Health applications throughout all sections while condensing unrelated material.
- The discussion of One Health primarily describes Y. pestis transmission dynamics across animals, humans, and environments but lacks actionable strategies. The few implementation suggestions presented remain overly theoretical (e.g., "improve surveillance" without methodological details). The authors should enhance practical relevance, for example, incorporating case studies of successful One Health interventions and proposing specific, measurable frameworks for integrated surveillance or control programs.
- Future projections currently rely on qualitative extrapolations of historical trends. To increase scientific rigor, I suggest the authors to integrate some quantitative modeling approaches.
- All tables appear to be inserted as low-resolution images.
Author Response
We thank you for the thorough and constructive feedback. We have carefully considered the major issues highlighted and have revised the manuscript accordingly to address these points, ensuring improved clarity, focus, and alignment with the One Health framework.
According to your comment we modified the following sentences to:
- Comment 1: The current emphasis on the One Health approach in the title and introduction is not sufficiently reflected in the article's content structure. While section 2 addresses this framework, other substantial portions (virulence factors, Yersinia pestis evolution, clinical features, and treatment/prevention) remain disconnected from this central theme. To strengthen coherence, the authors should either revise the title and introductory statements to better match the actual content distribution, or significantly expand One Health applications throughout all sections while condensing unrelated material.
Response 1: We thank you for this comment. We have carefully revised the manuscript to ensure that the One Health perspective is consistently integrated throughout the text in particular in the virulence factors, Yersinia pestis evolution portions. However for clinical features, and treatment/prevention we did consider them outside the focus of the aims of the review.
- Comment 2: The discussion of One Health primarily describes Y. pestis transmission dynamics across animals, humans, and environments but lacks actionable strategies. The few implementation suggestions presented remain overly theoretical (e.g., "improve surveillance" without methodological details). The authors should enhance practical relevance, for example, incorporating case studies of successful One Health interventions and proposing specific, measurable frameworks for integrated surveillance or control programs.
Response 2: Thank you for your suggestion. We have inserted in section 3 examples to improve practical relevance.
- Comment 3: Future projections currently rely on qualitative extrapolations of historical trends. To increase scientific rigor, I suggest the authors integrate some quantitative modeling approaches.
Response 3: Thank you to the reviewer for this suggestion. In the revised manuscript, we have added a specific section “2.4. Plague impact assessment and prediction” where we explicitly discuss qualitative and quantitative modeling approaches to assess and predict the impact of plague outbreaks.
- Comment 4: All tables appear to be inserted as low-resolution images.
Response 4: According to the comment, all tables have now been inserted following the MDPI instructions, using the ‘Table’ option in Microsoft Word, rather than as images.
Along with the cover letter, we submitted:
The final revised manuscript: biomedicines-3767536_resubmission_Round1.docx, containing the fully final revised text formatted according to journal guidelines.
The annotated version: Biomedicines_3767536_tracking.pdf, containing the original text and all modifications: the original text is shown in black, the newly added or modified text is highlighted in green and the deleted text is shown as strikethrough. This annotated version is provided solely to facilitate the editorial and peer review process and is not intended for publication. Please see the attachment.
Best Regards

Reviewer 5 Report
Comments and Suggestions for Authors
This is a nicely written and most comprehensive review on numerous aspects of plague, the devastating disease in past and remaing endemic in several regions. The authors have considered numerous aspects related to plague under the umbrella of One Health Approach and analyzed numerous factors implicated in transmission, survival, and pathogenicity of Yersinia pestis, the causative agent. The detailed survey of microbiology, virulence factors, and the measures to combat this bacterium is very impressive. I have no doubts that this review deserves publishing in Biomedicines; however, it is very large for a review in a scientific journal and resembles a nice chapter for a book. Hopefully, the Editors and Reviewers will endorse (have already endorsed) publishing of the whole ms. If there are limitations, this ms can be divided in two reviews, with the second one starting from Section 5.
Author Response
- Comment 1: This is a nicely written and most comprehensive review on numerous aspects of plague, the devastating disease in past and remaing endemic in several regions. The authors have considered numerous aspects related to plague under the umbrella of One Health Approach and analyzed numerous factors implicated in transmission, survival, and pathogenicity of Yersinia pestis, the causative agent. The detailed survey of microbiology, virulence factors, and the measures to combat this bacterium is very impressive. I have no doubts that this review deserves publishing in Biomedicines; however, it is very large for a review in a scientific journal and resembles a nice chapter for a book. Hopefully, the Editors and Reviewers will endorse (have already endorsed) publishing of the whole ms. If there are limitations, this ms can be divided in two reviews, with the second one starting from Section 5.
Response 1: We thank the Reviewer for the comments and positive evaluation of our manuscript. We appreciate that the reviewer recognized our effort to address the numerous aspects related to plague within the One Health framework. We have seriously considered the suggestions, as well as the ones of the other reviewers, including the advice to divide the manuscript into two reviews. Taking into account the additional recommendation to reduce some sections, we have worked to make certain parts more concise while keeping them informative. However, the editors and reviewers had endorsed publishing the whole manuscript. Therefore, we decided to proceed with publishing the manuscript as a single, integrated review. We hope the reviewers will appreciate the revised version of the manuscript and that it will be favorably considered for publication.
Along with the cover letter, we submitted:
The final revised manuscript: biomedicines-3767536_resubmission_Round1.docx, containing the fully final revised text formatted according to journal guidelines.
The annotated version: Biomedicines_3767536_tracking.pdf, containing the original text and all modifications: the original text is shown in black, the newly added or modified text is highlighted in green and the deleted text is shown as strikethrough. This annotated version is provided solely to facilitate the editorial and peer review process and is not intended for publication. Please see the attachment.
Best Regards

Round 2
Reviewer 3 Report
Comments and Suggestions for Authors
This manuscript is much improved and all of my concerns have been addressed. It is more focused and issues with its grammar have been rectified.